# Collagen-producing lung cell atlas identifies multiple subsets with distinct localization and relevance to fibrosis

Tatsuya Tsukui [1], Kai-Hui Sun[1], Joseph B. Wetter[2], John R. Wilson-Kanamori[3], Lisa A. Hazelwood[2], Neil C. Henderson [3], Taylor S. Adams[4], Jonas C. Schupp [4], Sergio D. Poli[5], Ivan O. Rosas[5], Naftali Kaminski [4], Michael A. Matthay[6], Paul J. Wolters[7] & Dean Sheppard [1✉]

Collagen-producing cells maintain the complex architecture of the lung and drive pathologic scarring in pulmonary fibrosis. Here we perform single-cell RNA-sequencing to identify all collagen-producing cells in normal and fibrotic lungs. We characterize multiple collagen-producing subpopulations with distinct anatomical localizations in different compartments of murine lungs. One subpopulation, characterized by expression of *Cthrc1* (collagen triple helix repeat containing 1), emerges in fibrotic lungs and expresses the highest levels of collagens. Single-cell RNA-sequencing of human lungs, including those from idiopathic pulmonary fibrosis and scleroderma patients, demonstrate similar heterogeneity and *CTHRC1*-expressing fibroblasts present uniquely in fibrotic lungs. Immunostaining and in situ hybridization show that these cells are concentrated within fibroblastic foci. We purify collagen-producing subpopulations and find disease-relevant phenotypes of *Cthrc1*-expressing fibroblasts in in vitro and adoptive transfer experiments. Our atlas of collagen-producing cells provides a roadmap for studying the roles of these unique populations in homeostasis and pathologic fibrosis.

[1] Lung Biology Center, Department of Medicine, University of California, San Francisco, San Francisco, CA, USA. [2] Abbvie Inc, North Chicago, IL, USA. [3] Centre for Inflammation Research, The Queen's Medical Research Institute, University of Edinburgh, Edinburgh, UK. [4] Section of Pulmonary, Critical Care and Sleep Medicine, Yale School of Medicine, New Haven, CT, USA. [5] Brigham and Women's Hospital, Harvard Medical School, Boston, MA, USA. [6] Departments of Medicine and Anesthesia, Cardiovascular Research Institute, University of California, San Francisco, San Francisco, CA, USA. [7] Division of Pulmonary, Critical Care, Allergy and Sleep Medicine, Department of Medicine, University of California, San Francisco, San Francisco, CA, USA. ✉email: Dean.Sheppard@ucsf.edu

The lung is a complex organ with marked regional differences in structure and function. Important sub-structures in the lung include the conducting airways, responsible for filtering and delivering inhaled and exhaled air, the bronchovascular bundles, conduits for air, blood, and lymphatics, and the alveolar region, responsible for gas exchange[1]. Collagens play central roles in maintaining the organization and structural integrity of these distinct anatomic features[2]. Perturbation of collagen homeostasis is an important contributor to common lung diseases, including pulmonary fibrosis. However, the cells responsible for normal collagen production in each of these regions and for excess accumulation of collagen in fibrotic lung diseases have not been well characterized. It has been widely assumed that there is a single cell type, often called a myofibroblast and characterized by expression of the contractile protein alpha-smooth muscle actin (α-SMA), that is the major collagen-producing cell in the lung. However, previous work by ourselves and others suggest that α-SMA is an inconsistent marker of collagen-producing cells in in normal murine lungs and in murine models of pulmonary fibrosis[3,4]. To enhance progress in understanding the cellular and molecular mechanisms underlying normal collagen homeostasis and fibrotic lung diseases, a better understanding of the cells responsible for collagen production is urgently needed.

The development of single-cell RNA-sequencing (scRNA-seq) allows us to evaluate the heterogeneity of tissue cell types with markedly improved precision. Several recently published studies have used this approach to characterize some of the mesenchymal populations in the lung. Zepp et al. identified mesenchymal alveolar niche cells and myogenic progenitor cells by using Axin2-CreER/Pdgfra-GFP/Wnt2-CreER reporter mice[5]. Xie et al. collected mesenchymal populations more broadly by sorting CD31, CD45, and Epcam-negative cells, and described 6–7 mesenchymal populations in normal and fibrotic murine lungs[6]. Another recent study performed scRNA-seq on all cells obtained with a single digestion protocol from fibrotic human lungs, but the overwhelming majority of cells obtained and analyzed were leukocytes and epithelial cells, with very few cells that were characterized as fibroblasts[7]. Overall, no studies to date have specifically focused on the cells responsible for production of collagen and other extracellular matrix (ECM) proteins and the anatomic location of these cell types in normal and fibrotic lungs.

Here, we perform scRNA-seq of cells obtained with a protocol designed to capture all of the collagen-producing cells from normal and fibrotic mouse and human lungs. We use gene expression signatures to classify fibroblast subpopulations based on their anatomical localizations, and developed fluorescence-activated cell sorting (FACS) strategies to purify those populations. Through this approach, we identify molecularly distinct populations of fibroblasts that populate distinct anatomic locations, in the walls of conducting airways (peribronchial), surrounding bronchovascular bundles (adventitial) and embedded within the alveolar regions of the lung (alveolar). We also identify a unique population of Cthrc1 (collagen triple helix repeat containing 1)+ fibroblasts, which are mostly found in fibrotic lungs in both mice and humans and expresses the highest levels of type 1 collagen and other ECM genes. Purified Cthrc1+ fibroblasts are more migratory than other subsets of collagen-producing cells and demonstrate an enhanced capacity to colonize the lungs of bleomycin-treated mice. These findings identify the distinct gene and cell surface protein expression patterns that characterize fibroblast subsets with distinct anatomic localizations. The Cthrc1+ subset we describe is likely to play an important role in the development of pulmonary fibrosis in mice and humans.

## Results

**scRNA-seq of murine lung cells in normal and fibrotic lungs.** We used Col1a1-EGFP (Col-GFP) reporter mice to harvest all of the collagen-producing cells in murine lungs[8]. We induced lung fibrosis by intratracheally instilling bleomycin into two Col-GFP mice, and prepared a single cell suspension from the lungs 14 days after the bleomycin treatment. Two untreated Col-GFP mice were used as control. In addition to GFP+ cells, we sorted GFP− cells to compare gene expression patterns in the cells responsible for collagen production to pattern seen in other cell types. We performed scRNA-seq of each sample using the 10x Genomics platform (Fig. 1a). After quality filtering, we obtained 25953 cell profiles from all the GFP+ and GFP− samples. We performed dimensionality reduction with canonical correlation analysis (CCA) subspace alignment and performed unsupervised clustering (Fig. 1b)[9]. Analysis of representative markers identified clusters of endothelial, epithelial, and mesothelial cells as well as hematopoietic cells including macrophages, monocytes, neutrophils, dendritic cells, natural killer cells, and lymphocytes (Fig. 1c, Supplementary Fig. 1a). To characterize the heterogeneity of collagen-producing cells in higher resolution, we focused on the cells from GFP+ samples and removed all clusters with lineage markers, which did not consistently express Col1a1 except a small cluster of mesothelial cells (Fig. 1c). Re-clustering of Col1a1+ cells revealed 12 clusters from 12,855 cells (Fig. 1d). All the clusters included cells from both bleomycin-treated and untreated lungs except clusters 8 and 11, which were mostly from bleomycin-treated lungs (Fig. 1e, Supplementary Fig. 1b). The clusters were categorized into two superclusters: one composed of clusters 0, 1, 2, 4, 6, 8, 10 with higher Col1a1 expression, and the other composed of clusters 3, 5, 7, 9 with higher Acta2 expression (Fig. 1f). Cluster 11 is proliferating cells characterized by the expression of Mki67 and Cdc20 (Supplementary Fig. 1c). Clusters 5 and 7 expressed smooth muscle cell markers such as Acta2 and Myh11 (Fig. 1f, g). Cluster 9 expressed pericyte markers such as Mcam, Cspg4 and the highest level of Pdgfrb (Fig. 1g).

**Identification of fibroblast subsets in normal lungs.** To further characterize these populations, we performed differentially expressed gene analysis and identified expression markers of clusters of interest (Fig. 2a). Clusters 0, 1, 2, 10 shared common markers including Npnt and Ces1d (Fig. 2a). Slc7a10 is specifically expressed in cluster 0 (Fig. 2a). Clusters 4 and 6 shared some markers such as Pi16 and Dcn (Fig. 2a). Cluster 4 uniquely expressed cytokines such as Il33 and Ccl11, while cluster 6 uniquely expressed Adh7 (Fig. 2a). Cluster 3 highly expressed Hhip, Aspn and Fgf18 (Fig. 2a).

To examine the localization of these clusters in vivo, we used proximity ligation in situ hybridization (PLISH), a recently-reported in situ hybridization technology with high sensitivity and specificity[10]. The PLISH signal is observed as bright dots, which represent the cellular locations of target mRNA. In normal lungs, Col-GFP+ cells were mainly detected at alveolar walls and adventitial cuffs surrounding large airways and arteries in bronchovascular bundles (Supplementary Fig. 2a). We found that Col-GFP+ cells in alveoli expressed both Npnt and Ces1d, markers for clusters 0, 1, 2, 10 (Fig. 2b). We also detected Ces1d signals in airway epithelial cells, which is consistent with our whole lung scRNA-seq data (Supplementary Fig. 2b), but not in Col-GFP+ cells in bronchovascular cuffs (Fig. 2b). Among these alveolar fibroblast clusters, cluster 0 was most prominent in the lungs of untreated mice (Fig. 1e, Supplementary Fig. 1b). In contrast, Pi16 was expressed by Col-GFP+ cells in the cuffs (Fig. 2c). Adh7, a marker for cluster 6, was also detected in Col-GFP+ cells in the cuffs (Fig. 2c). Furthermore, signals for Ccl11

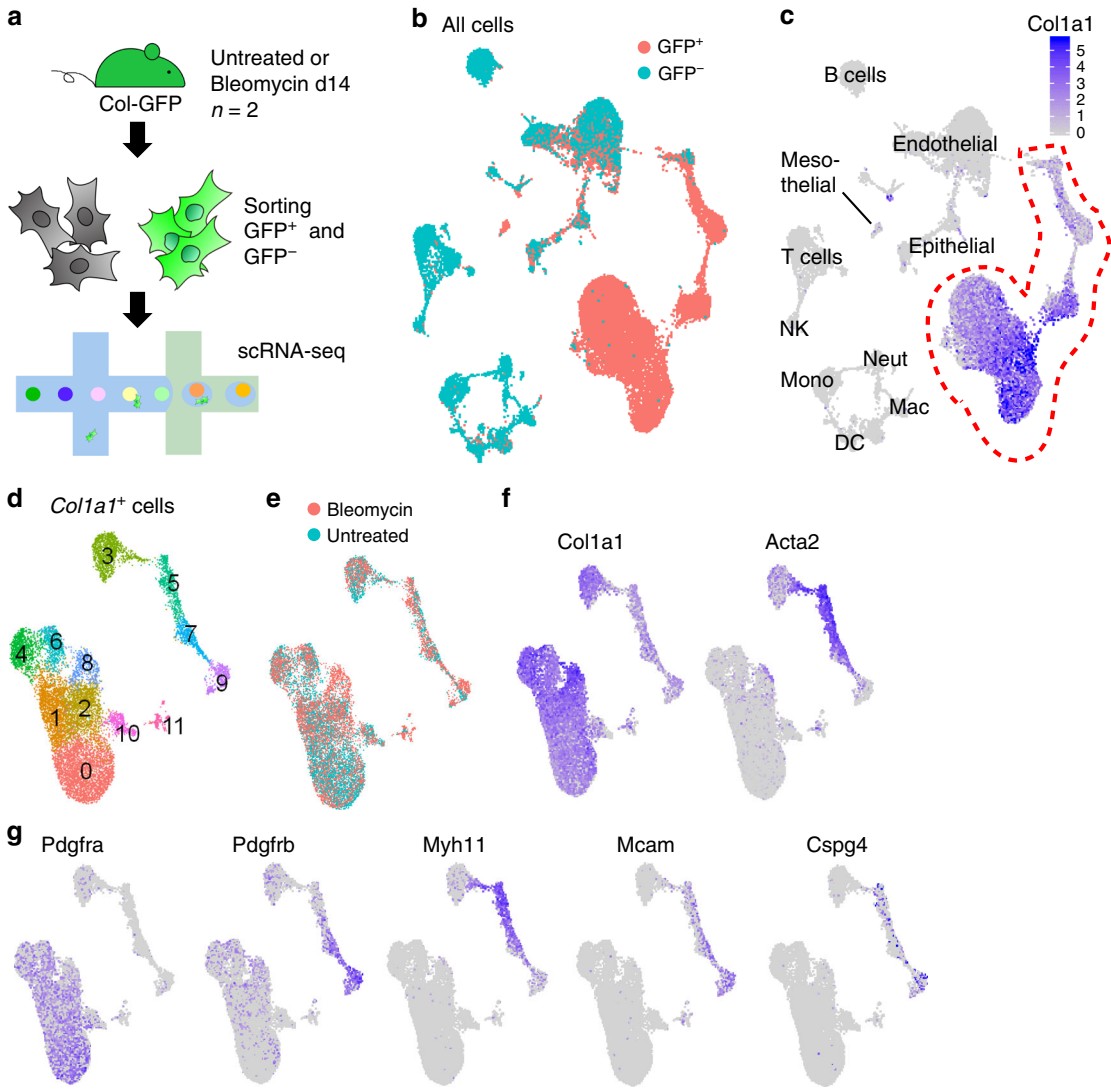

**Fig. 1 scRNA-seq of murine lung cells in normal and fibrotic lungs. a** Schematic of scRNA-seq sample preparation. **b** Uniform manifold approximation and projection (UMAP) plot of all cells colored by GFP+ and GFP− samples. **c** *Col1a1* expression on UMAP plot of all cells. See Supplementary Fig. 1a for identifying the lineages. NK, natural killer cell; Neut, neutrophil; Mac, macrophage; DC, dendritic cell; Mono, monocyte. **d–f** UMAP plots of *Col1a1*+ cells. **d** Unsupervised clustering identifies 12 clusters. **e** Cells from bleomycin-treated lung are shown in red. Cells from untreated lungs are shown in blue. **f, g** Gene expression levels for each gene.

were enriched in Col-GFP+ cells in the cuffs (Fig. 2d). These findings are consistent with a recent report, which identified *Il33*-expressing stromal cells in the adventitial cuffs[11], but also suggests that there is another kind of adventitial fibroblast characterized by *Adh7* expression that does not express cytokine genes.

A previous study identified *Hhip*+ mesenchymal cells around airways in the lung[12]. PLISH for *Hhip* and *Aspn*, markers for cluster 3, also showed signal enrichment in subepithelial Col-GFP+ cells around bronchi (Fig. 2e). Those *Hhip*+ Col-GFP+ sub-epithelial cells were distinct from airway smooth muscle cells characterized by *Actc1* expression (Supplementary Fig. 2c, d).

Three-dimensional imaging of cleared thick lung sections of Col-GFP mice revealed that those subepithelial Col-GFP+ cells were intercalated between airway smooth muscle cells localized just below the airway epithelium (Fig. 3a, b, Supplementary Movie 1). Type 4 collagen staining showed that subepithelial Col-GFP+ cells made contacts with epithelial basement membranes (Fig. 3c, Supplementary Movie 2). Adventitial fibroblasts closely associated with type 4 collagen surrounding the bronchovascular

cuffs (Fig. 3c, Supplementary Movie 2). A previous report showed that *Pdgfra*+ cells in alveolar walls have close association with type 2 alveolar epithelial cells (AEC2s)[13]. We crossed Col-GFP mice with Shh-Cre/Rosa26-lox-stop-lox-tdTomato mice, in which all epithelial cells in the lung express tdTomato[14], to investigate the interaction of Col-GFP+ cells and epithelial cells in alveoli. Consistent with the previous report, alveolar fibroblasts were located in close proximity to AEC2s (Fig. 3d, Supplementary Movie 3).

These results identify 4 subclusters of alveolar fibroblasts (clusters 0, 1, 2, 10), two types of adventitial fibroblasts (clusters 4, 6), and a cluster of peribronchial fibroblasts (clusters 3) (Fig. 3e). These fibroblast subpopulations have distinct localizations in different compartments of the lung (Fig. 3f).

**Population comparison to previous studies**. Cluster 3 also expressed *Lgr5* (Supplementary Fig. 3a), suggesting that peri-bronchial fibroblasts may correspond to the *Lgr5*+ mesenchymal cells reported by Lee et al.[15]. Lgr6 was expressed by cluster 3 and

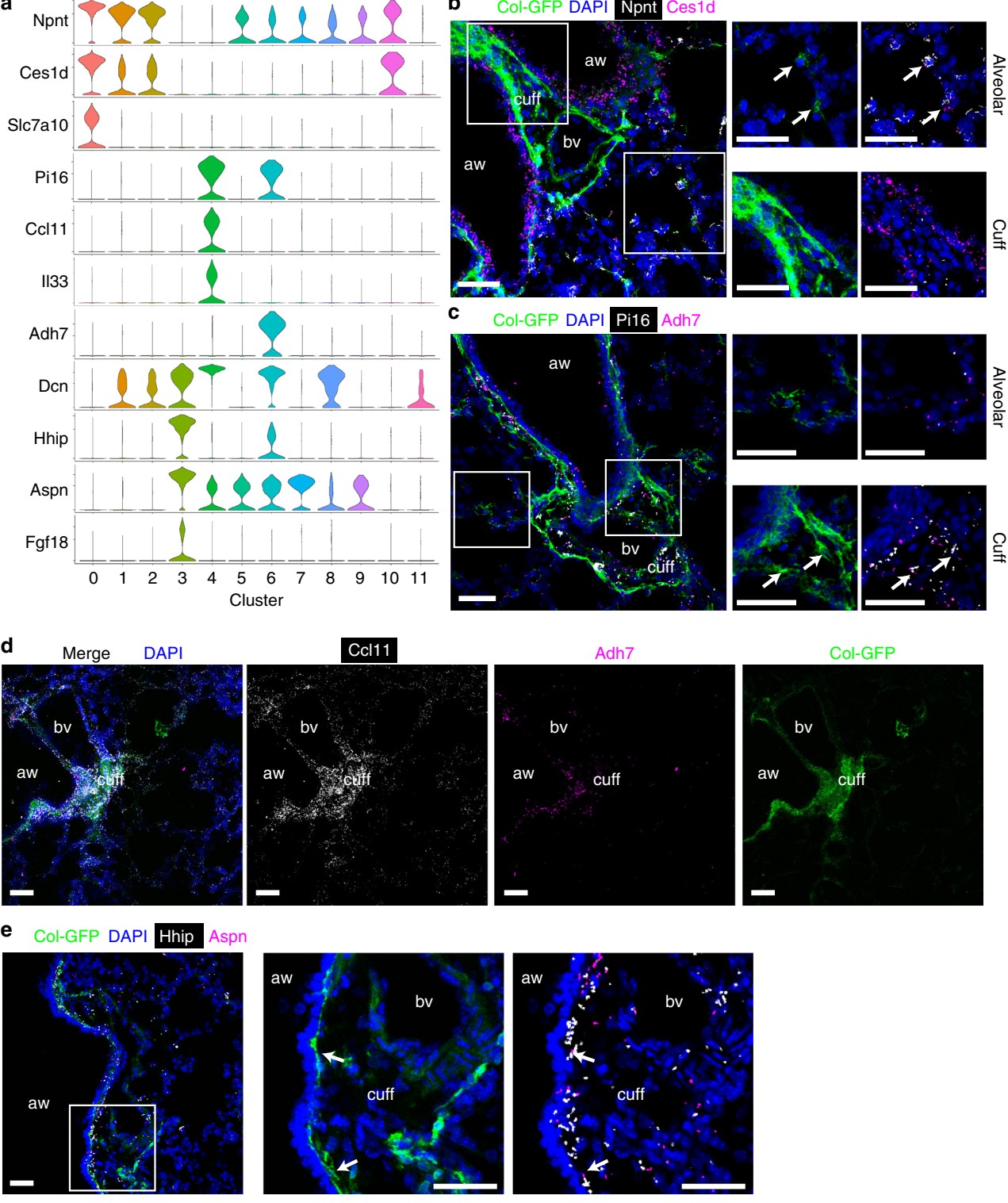

**Fig. 2 Identification of alveolar, adventitial, and peribronchial fibroblasts in untreated lungs. a** Violin plots showing the expression levels in each cluster of representative marker genes. **b**, **c** Proximity ligation in situ hybridization (PLISH) images for *Npnt* (white) and *Ces1d* (magenta) (**b**), or for *Pi16* (white) and *Adh7* (magenta) (**c**). Magnified images of the white squares are shown in right panels. Arrows indicate co-localization of PLISH signals in GFP+ cells. **d** PLISH images for *Ccl11* (white) and Adh7 (magenta). **e** PLISH images for *Hhip* (white) and *Aspn* (magenta). Magnified images of the white square are shown in right panels. Arrows indicate co-localization of PLISH signals in GFP+ cells. **b**–**e** Col-GFP is shown in green. DAPI signal is shown in blue. Scale bars, 50 μm. aw, airway; bv, blood vessel; cuff, cuff space. Images are representative of three experiments (n ≥ 2).

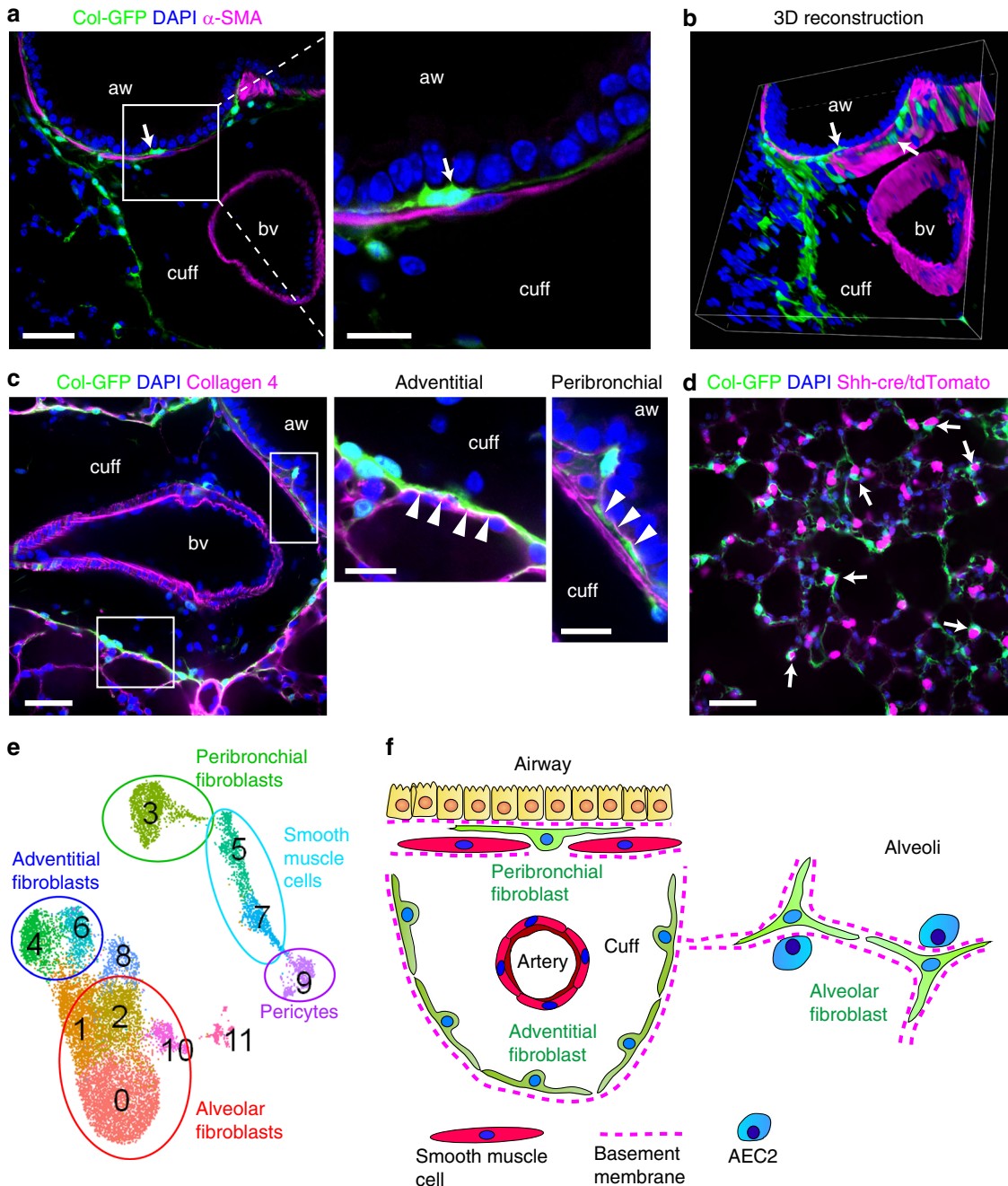

**Fig. 3 Characterization of alveolar, adventitial, and peribronchial fibroblasts. a–d** Cleared thick sections of untreated lungs were imaged by confocal microscopy. Col-GFP is shown in green. DAPI is shown in blue. aw, airway; bv, blood vessel; cuff, cuff space. Images are representative from two experiments ($n \geq 2$). **a, b** α-SMA staining is shown in magenta. Arrows indicate peribronchial fibroblasts. **a** Magnified image of the white square is shown in the right panel. Scale bars, 50 μm (left panel), 20 μm (right panel). **b** Series of z-stack images were 3D-reconstructed. See also Supplementary Movie 1. **c** Collagen 4 staining is shown in magenta. Magnified images of the white squares are shown in the right panels. Arrowheads indicate basement membranes. Scale bars, 50 μm (left panel), 20 μm (center and right panels). See also Supplementary Movie 2. **d** Representative image of Shh-Cre/Rosa26-lox-stop-lox-tdTomato/Col-GFP mice. tdTomato signal is shown in magenta. Arrows indicate the close localization of alveolar fibroblasts and AEC2s. Scale bars, 50 μm. See also Supplementary Movie 3. **e** Collagen-producing subpopulations identified are shown on UMAP plot of *Col1a1*+ cells. **f** Schematic showing the distinct localization of fibroblast subpopulations.

some of the smooth muscle cells (Supplementary Fig. 3a)[15]. Zepp et al. used *Axin2*, *Pdgfra*, and *Wnt2* to classify mesenchymal populations[5]. *Axin2* was broadly expressed in all mesenchymal populations in our data set (Supplementary Fig. 3b). *Wnt2* was mainly expressed in clusters 0, 1, 2 (Supplementary Fig. 3b). *Sfrp4* and *Wif1* were reported as markers for mesenchymal alveolar niche cells (MANC)[5]. Cluster 4 expressed *Sfrp4*, and clusters 3, 6

expressed *Wif1* in our data set (Supplementary Fig. 3b). Clusters 5, 7, 9, which are smooth muscle cells and pericytes, expressed the markers of *Axin2*+ myofibrogenic progenitor (AMP), *Notch3* and *Gucy1a3* (Supplementary Fig. 3b)[5]. Xie et al. reported *Col13a1* and *Col14a1* expressing fibroblasts[6]. Clusters 0, 1, 2 expressed *Col13a1*, and clusters 4, 6 expressed *Col14a1* (Supplementary Fig. 3c). The population described as "myofibroblasts" in Xie et al. has the same

markers as cluster 3, including *Hhip* and *Aspn* (Fig. 2a). *Higd1b* and *Cox4i2*, which are the markers of *Pdgfrb* hi populations in Xie et al., were expressed in pericytes (cluster 9), although we found pericytes from both untreated and bleomycin-treated lungs unlike Xie et al. (Fig. 1e, Supplementary Fig. 3c)[6]. This apparent difference could be explained by differences in the digestion protocols used. The specific markers for the cells described as "lipofibroblasts" in Xie et al. such as *Ear1*, *Ear2*, *Mrc1*, or *Clec4n*, were all expressed in *Ptprc* (CD45)+ *Itgax* (CD11c)+ *Siglecf*+ alveolar macrophages (Supplementary Fig. 1a, 3d)[6]. General lipofibroblast markers such as *Plin2* and *Lpl* were also expressed in clusters 0, 1, 2, consistent with previous reports that *Pdgfra*+ *Tcf21*+ *Col13a1*+ alveolar fibroblasts have some phenotypic features of lipofibroblasts (Supplementary Fig. 3e)[13,16].

***Cthrc1*+ cells express pathologic ECM genes in fibrotic lungs.** Next, we analyzed specific markers for cluster 8, which was mostly from bleomycin-treated lungs. Cluster 8 specifically expressed *Cthrc1*, which was previously shown to be increased in bulk RNA sequencing from the lungs of patients with idiopathic pulmonary fibrosis (IPF) (Fig. 4a)[17]. Other markers for cluster 8 include ECM-related genes such as *Postn*, *Spp1*, *Fn1*, and *Tnc* (Fig. 4a). *Col1a1* and *Col3a1* expression were also the highest in cluster 8 (Fig. 4a). Feature plots of *Col1a1* and *Cthrc1* showed the correlation of the two genes in bleomycin-treated lungs (Fig. 4b). *Cthrc1* was specific for the population with the highest collagen expression and not significantly detected in other lung cells, while the myofibroblast marker, *Acta2* was predominantly expressed by low collagen-expressing cells (Supplementary Fig. 4a). PLISH for *Cthrc1* and *Col1a1* in bleomycin-treated lung sections showed that *Cthrc1* was expressed at the periphery of clusters of *Col1a1*+ cells in the alveolar region (Fig. 4c). To computationally investigate the potential progenitors of *Cthrc1*+ fibroblasts, we performed RNA velocity analysis, in which the direction of cellular differentiation is predicted based on the reads from both spliced and unspliced RNA[18]. Vectors pointing into cluster 8 mostly came from cluster 2, which expresses multiple markers of alveolar fibroblasts (Fig. 4d). This analysis is consistent with the alveolar localization of *Cthrc1*+ fibroblasts. To further characterize the relevance of alveolar fibroblasts and cluster 8, we zoomed in on clusters 0, 1, 2, 8 (Fig. 4e), and performed pseudotime trajectory analysis[19]. The resulting trajectory proposed a differentiation from cells in cluster 0 into clusters 1 and 2, with cells in cluster 8 forming a terminal state at the end (Fig. 4f). These results suggest that *Cthrc1*+ fibroblasts are activated to produce pathologic ECM in fibrotic lesions and may principally differentiate from alveolar fibroblasts.

Interestingly, we found that the expression of alveolar fibroblast markers, such as *Pdgfra*, *Npnt*, *Ces1d*, and *Tcf21* gradually decreased along the pseudotime towards cluster 8 (Fig. 4g, h, Supplementary Fig. 4b), while ECM genes such as *Col1a1*, *Col3a1*, and *Col5a1* gradually increased along the pseudotime (Fig. 4g, h, Supplementary Fig. 4b). These results may indicate the differentiation of alveolar fibroblasts into functionally distinct ECM-producing cells.

Peyser et al. suggested that activated fibroblasts were not uniquely defined from scRNA-seq of bleomycin-treated mouse lungs[20]. To clarify the discrepancy to our data, we analyzed their scRNA-seq data (GSE129605). After focusing on *Col1a1*+ cells, we found that a fraction of *Col1a1*+ cells, that expressed the highest levels of collagen and were mainly from bleomycin-treated lungs, grouped together and expressed cluster 8 markers such as *Cthrc1*, *Tnc*, *Spp1*, and *Fst*, consistent with our findings (Supplementary Fig. 4c). These pathologic fibroblasts were not bioinformatically identified as a distinct cluster possibly because there were

relatively small number of *Col1a1*+ cells (~2000 cells)[21]. It is also possible that the dissociation conditions used in Peyser et al. did not capture all of these cells.

**scRNA-seq of normal and fibrotic lungs of human.** Next, we performed scRNA-seq of human lung cells to explore the heterogeneity of collagen-producing cells in normal and fibrotic lungs. We collected samples from explanted lungs of three IPF patients and two scleroderma patients, which allowed us to compare pathologic collagen-producing populations in two fibrotic diseases with different etiology. Three normal lung samples were collected from rejected lung transplant donors. We sorted lineage (CD31, CD45, CD235a, EPCAM)− cells to enrich mesenchymal cells. We also collected CD235a (erythrocyte marker)− cells as all lung cells. All the cell profiles were aggregated with mutual nearest neighbors (MNN) batch correction, which is effective for correcting diverse batch-to-batch differences (Fig. 5a)[22]. We obtained 83316 cells after quality filtering. By analyzing representative markers, we identified T cells, B cells, dendritic cells, epithelial cells, endothelial cells, macrophages, and *COL1A1*+ mesenchymal cells (Fig. 5b, Supplementary Fig. 5a). To characterize the heterogeneity of collagen-producing cell populations, we focused on lineage− samples and re-clustered *COL1A1*+ cells (Fig. 5c). Unsupervised clustering identified 7 clusters from 48,587 cells (Fig. 5d). Cell number of each cluster from each disease is shown in Supplementary Fig. 5b. Clusters 1, 3, 4 expressed *PDGFRA*, and clusters 0, 2, 6 expressed *PDGFRB* (Fig. 5e). Cluster 0 expressed pericyte markers including *RGS5* and *MCAM*, and clusters 2, 6 expressed the highest levels of smooth muscle markers including *MYH11* and *MYL9* (Fig. 5e). We observed increased number of pericytes (cluster 0) both in IPF and scleroderma lungs (Supplementary Fig. 5b). Cluster 1, which was enriched with cells from normal lungs, expressed alveolar fibroblast markers such as *NPNT* and *CES1* (Fig. 5e, Supplementary Fig. 5b). Cluster 4 expressed the markers of adventitial fibroblasts such as *PI16 and DCN* (Fig. 5e). Cluster 5 markers include *FGF18* and *WIF1*, which were also expressed in peribronchial fibroblasts in mice (Fig. 5e). Cluster 3 showed the highest *COL1A1* expression and was enriched with cells from fibrosis patients (Fig. 5c, f, Supplementary Fig. 5b). As seen in mice, *CTHRC1* expression correlated with *COL1A1* expression and was largely restricted to cells from fibrotic lungs, and *ACTA2* expression was poorly correlated with *COL1A1* (Fig. 5f). These findings were consistent across all the samples we obtained (Supplementary Fig. 5c). As we found in mice, *CTHRC1*+ cells also expressed other ECM genes such as *TNC*, *POSTN*, and *COL3A1* (Fig. 5g).

To examine the consistency of our findings in larger data sets, we investigated scRNA-seq data from 29 normal and 32 IPF lungs[23]. After focusing on *COL1A1*+ cells, we found similar heterogeneous populations including *RGS5*+ *CSPG4*+ pericytes, *MYH11*+ *MYL9*+ smooth muscle cells, *NPNT*+ *CES1*+ alveolar fibroblasts, and *PI16*+ *DCN*+ adventitial fibroblasts (Supplementary Fig. 6a, b). A fraction of cells, which were mainly from IPF lungs, highly expressed *CTHRC1* and other ECM-related genes such as *COL1A1*, *COL3A1*, *TNC*, and *POSTN* (Supplementary Fig. 6c). Emergence of *CTHRC1*+ fibroblasts in lungs of scleroderma patients was also reported in a recent scRNA-seq study[24]. Moreover, we analyzed scRNA-seq data (GSE128033) from Morse et al.[25], in which the authors collected samples from 3 normal and 3 IPF lungs. We found that, consistent with our findings, a fraction of *COL1A1*+ cells expressed *CTHRC1* and other ECM-related genes but not the highest level of *ACTA2* (Supplementary Fig. 6d).

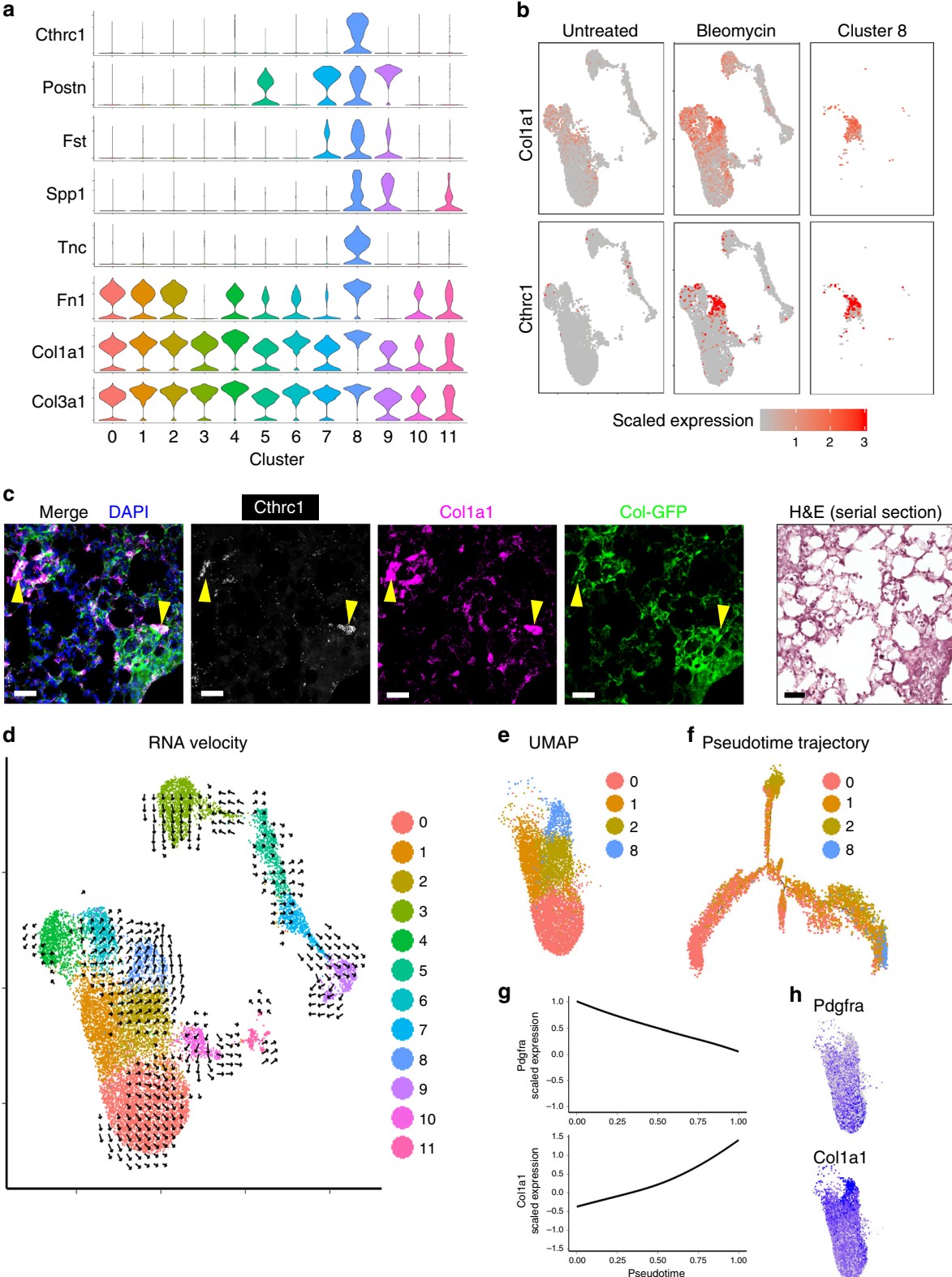

**Fig. 4 Cthrc1⁺ cells express pathologic ECM genes in fibrotic lungs. a** Violin plots showing markers of cluster 8. **b** Expression levels of *Col1a1* and *Cthrc1* on UMAP plots. Cells from untreated lungs, bleomycin-treated lungs, or cluster 8 are shown separately in each column. **c** PLISH images for *Cthrc1* (white) or *Col1a1* (magenta) 14 days after bleomycin treatment. Col-GFP is shown in green. DAPI signal is shown in blue. H&E staining of serial section is also shown. Arrowheads indicate co-localization of *Cthrc1* and *Col1a1* in GFP⁺ cells. Scale bar, 50 μm. Images are representative from three experiments (n ≥ 2). **d** RNA velocity analysis overlaid on UMAP plot. Direction indicates transition towards the estimated future state of a cell. **e** UMAP plot showing only clusters 0, 1, 2, and 8. **f** Pseudotime trajectory analysis by Monocle 2. **g** Expression levels of *Pdgfra* and *Col1a1* were analyzed along the pseudotime and fitted cubic smoothing spline curves are shown. The y-axis is scaled gene expression level. **h** *Pdgfra* and *Col1a1* expression levels on UMAP plots.

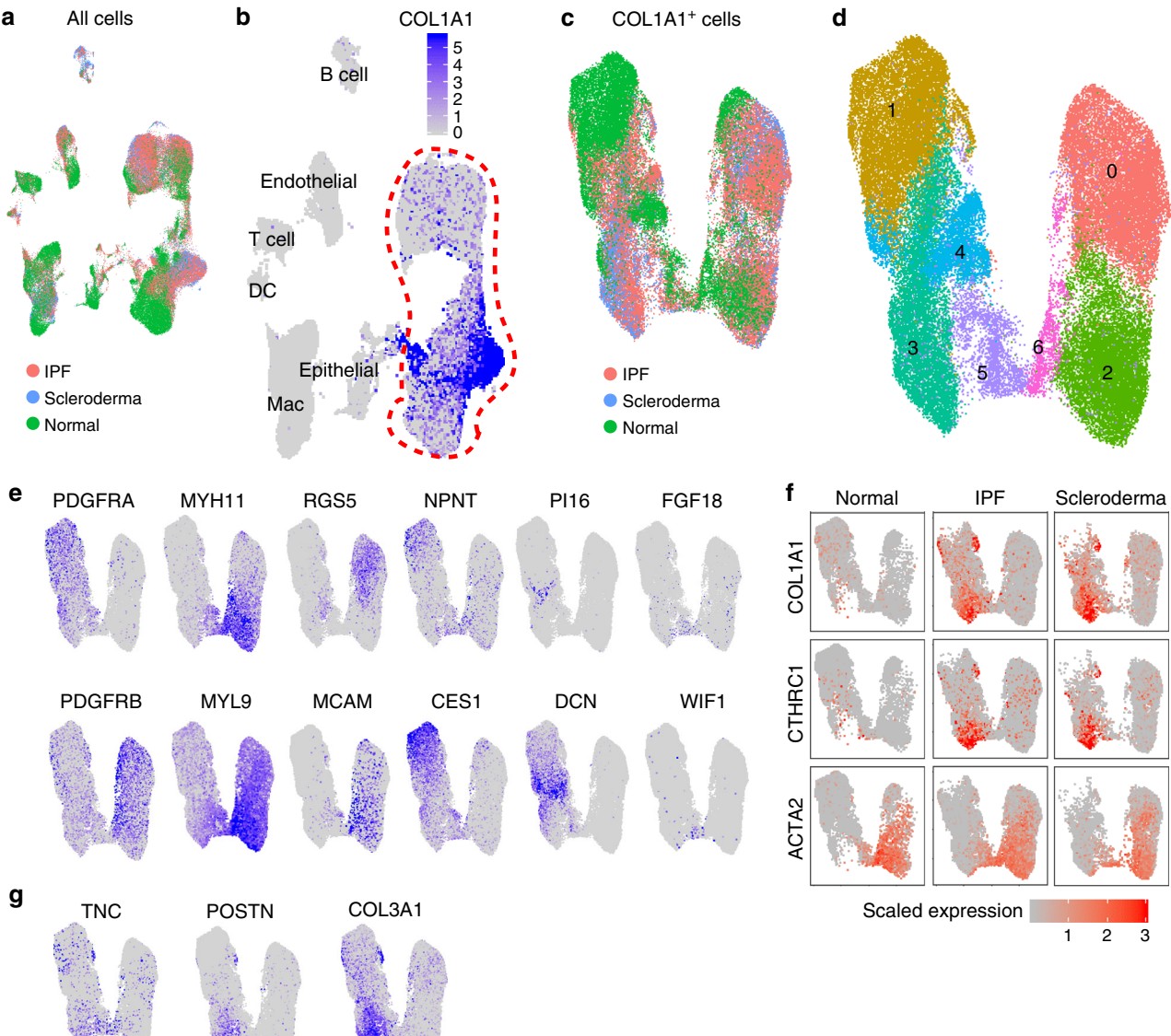

**Fig. 5 scRNA-seq identifies CTHRC1⁺ pathologic fibroblasts in human fibrotic lungs. a** UMAP plot of all cells acquired by scRNA-seq of human samples. Cells were obtained from three normal lungs (green), three IPF lungs (red), and two scleroderma lungs (blue). **b** *COL1A1* expression on UMAP plot. See Supplementary Fig. 5a for lineage identification. DC, dendritic cells; Mac, macrophage. **c** UMAP plot of COL1A1⁺ cells. **d** Unsupervised clustering identified 7 clusters. **e** Expression levels of representative genes are shown on UMAP plots. **f** Expression of *COL1A1*, *CTHRC1*, and *ACTA2* are shown on UMAP plots. Cells from normal, IPF, and scleroderma lungs are shown separately in each column. **g** Expression levels of selected ECM genes on UMAP plots.

To localize pathologic fibroblasts in fibrotic lungs, we performed RNAScope in situ hybridization (ISH) and antibody staining for *CTHRC1* in IPF lung sections (Fig. 6a). The highest *CTHRC1* RNA and protein expression was observed within fibroblastic foci, which are known as active sites of collagen production and leading edges of fibrogenesis (Fig. 6a)[26,27]. These results support the idea that *CTHRC1*⁺ fibroblasts could contribute to pathologic fibrogenesis in IPF. Previous reports showed that fibroblastic foci in IPF are α-SMA-positive[26]. To investigate the relation between CTHRC1⁺ pathologic fibroblasts and α-SMA⁺ cells, we stained sequential sections of IPF lungs for these markers (Fig. 6b). Although fibroblastic foci populated with CTHRC1⁺ pathologic fibroblasts were also positive for α-SMA, α-SMA⁺ cells were also evident outside fibroblastic foci (Fig. 6b). Taken together with our scRNA-seq data showing the higher ACTA2 expression in smooth muscle cells and pericytes, these data suggest that α-SMA is not a specific marker of pathologic fibroblasts producing the highest levels of ECM.

**FACS purification of fibroblast subsets.** One of the challenges in fibrosis research is a lack of methodology to purify mesenchymal subpopulations. To establish the methodology in mouse, we sought surface markers specific for each of the collagen-producing subpopulations. *Pdgfra* was widely expressed by alveolar and adventitial fibroblasts, although *Cthrc1*⁺ cluster 8 expressed relatively low levels of *Pdgfra* (Fig. 1g). We found that adventitial but not alveolar fibroblasts expressed *Ly6a* (Fig. 7a). Peribronchial fibroblasts were negative for *Pdgfra* but highly expressed *Cd9* (Fig. 7a). These results suggest that alveolar, adventitial, and peribronchial fibroblasts can be distinguished by the expressions of *Pdgfra*, *Ly6a*, and *Cd9*, while smooth muscle cells and pericytes can be distinguished by *Mcam* (Fig. 7b). To confirm this, we stained lung cells from untreated Col-GFP mice for these markers and performed sorting via FACS. After gating dump (CD31, CD45, EpCAM, Ter119)⁻ and GFP⁺ cells, we purified Mcam⁺ cells, Mcam⁻ Sca1 (Ly6a)⁺ cells as adventitial fibroblasts, Mcam⁻ Sca1⁻ Pdgfra⁺ cells as alveolar fibroblasts,

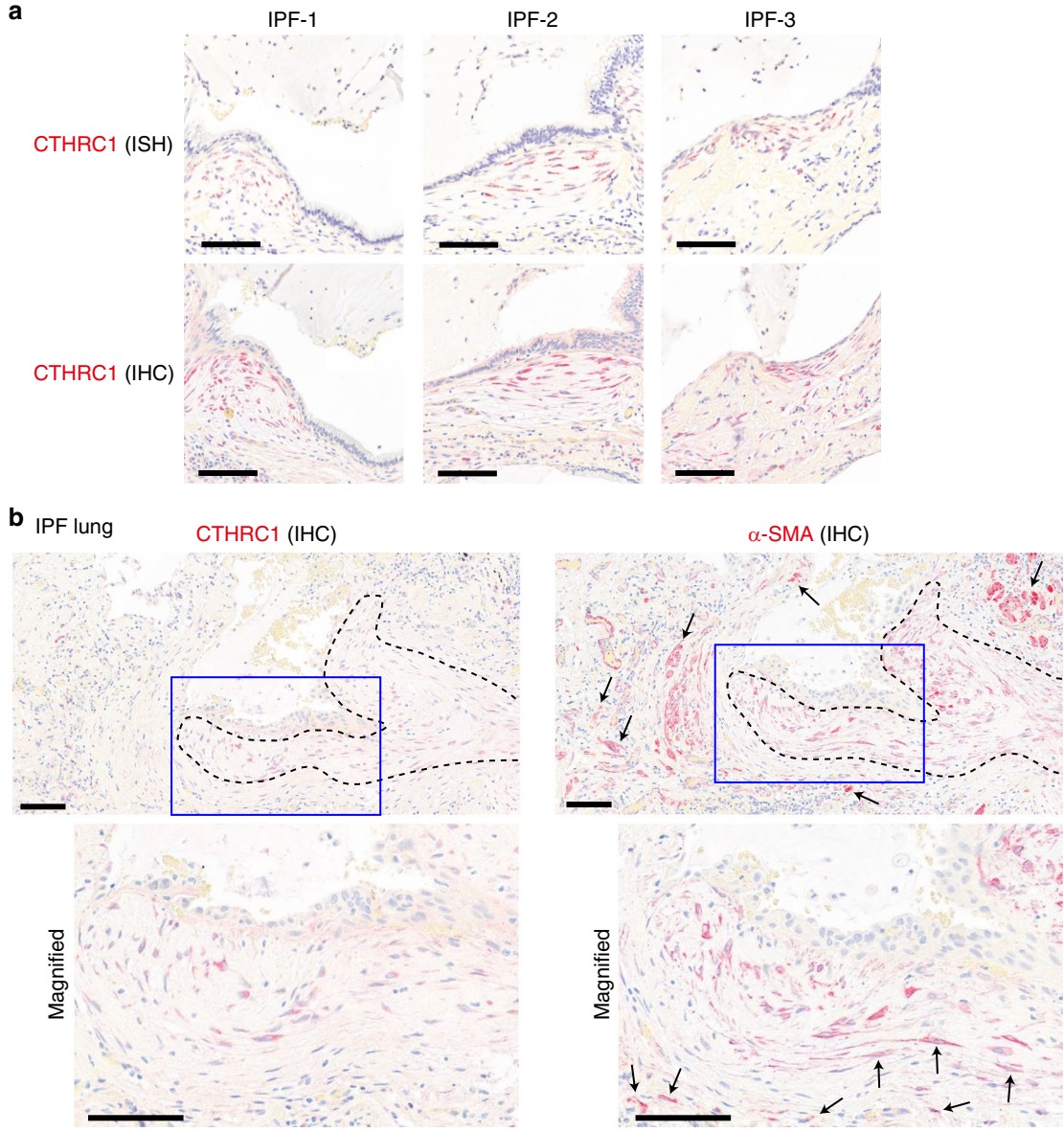

**Fig. 6 CTHRC1+ pathologic fibroblasts are localized within fibroblastic foci in IPF. a** Representative images of in situ hybridization (ISH) and immunohistochemistry (IHC) for CTHRC1 in the sections from three IPF patients. Scale bars, 100 μm. **b** CTHRC1 and α-SMA antibody staining in sequential sections of IPF lungs. Fibroblastic foci are outlined by dotted lines. Areas inside blue squares are magnified in lower panels. Arrows indicate α-SMA+ cells outside CTHRC1+ areas. Images are representative from four IPF patients. Scale bars, 100 μm.

and Mcam− Sca1− Pdgfra− CD9+ cells as peribronchial fibroblasts (Fig. 7c). All GFP+ cells were also sorted for comparison. We checked the expression of specific markers in sorted cells by quantitative Real Time PCR (qPCR) analysis. As expected, alveolar fibroblasts expressed *Slc7a10*, *Npnt*, and *Ces1d* (Fig. 7d, Supplementary Fig. 7a). Adventitial fibroblasts expressed *Pi16* (Fig. 7d). *Il33* and *Adh7* expressions were also enriched in adventitial fibroblasts, suggesting that the Sca1+ gating collected both clusters 4 and 6 (Supplementary Fig. 7a). Peribronchial fibroblasts expressed *Hhip* and *Aspn* (Fig. 7d, Supplementary Fig. 7a). Mcam+ cells highly expressed *Pdgfrb* and *Acta2*, which are the markers for pericytes and smooth muscle cells (Fig. 7d, Supplementary Fig. 7a). These results suggest that alveolar, adventitial, peribronchial fibroblasts and Mcam+ smooth muscle cells and pericytes can be purified based on Pdgfra, Sca1, and CD9 expressions in Col-GFP mice.

A previous study showed that bulk GFP+ cells from uninjured Col-GFP mice can colonize the lungs of bleomycin-treated, but not untreated mice after intratracheal adoptive transfer[28]. We used this transfer model to determine which of the subpopulations we identified have this capacity. We purified alveolar fibroblasts, adventitial fibroblasts, peribronchial fibroblasts, and Mcam+ cells from untreated Col-GFP mice, and transferred identical numbers of each subpopulation into bleomycin-treated wild type mice (Fig. 7e). Four days post-transfer, we harvested host lungs and counted GFP+ cells via flow cytometry. Among the four populations, both adventitial fibroblasts and alveolar fibroblasts showed engraftment potential, with adventitial fibroblasts colonizing modestly better than alveolar fibroblasts (Fig. 7f). The engraftment potential of peribronchial fibroblasts was about half that of alveolar fibroblasts (Fig. 7f). Mcam+ cells showed minimal engraftment potential, confirming our previously

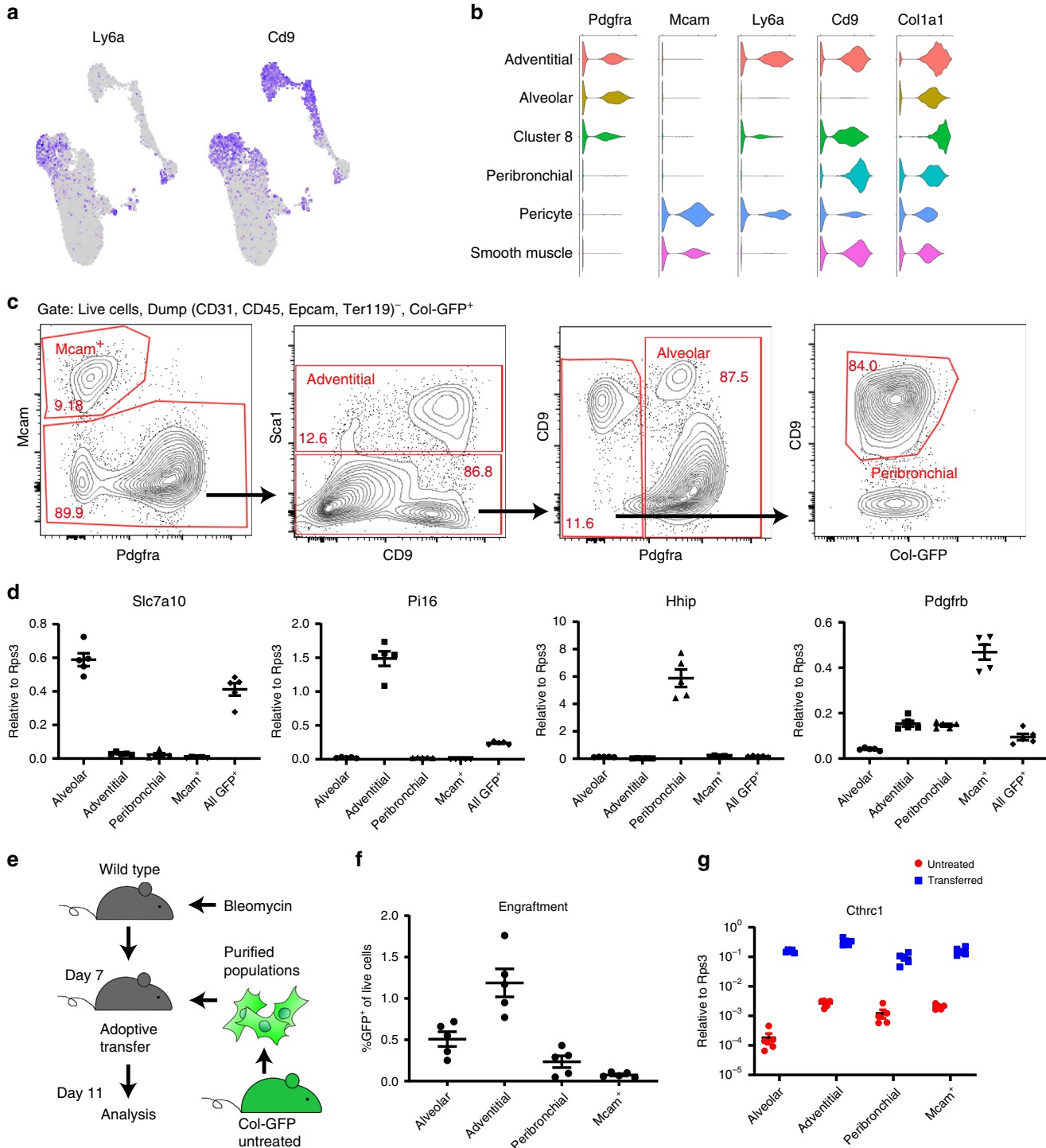

**Fig. 7 FACS purification of fibroblast subpopulations and their ability to colonize fibrosing lungs in adoptive transfer. a** Expression levels of *Ly6a* and *Cd9* on UMAP plots from murine data. **b** Violin plots showing expressions of surface markers on collagen-producing populations. **c** Gating strategy to purify Mcam⁺ cells, adventitial fibroblasts, alveolar fibroblasts, and peribronchial fibroblasts from untreated lungs. **d** qPCR analysis of purified cells from untreated lungs. $n = 5$ mice. **e** Schematic for the adoptive transfer experiment. **f** The number of GFP⁺ cells in host lungs were analyzed. $n = 5$ mice. **g** qPCR analysis of purified GFP⁺ cells from the host lungs. Cells purified from untreated lungs were used as control. $n = 5$ mice. **c**, **d**, **f**, **g** Data are representative from two experiments. Source data are provided as a Source Data file. **d**, **f**, **g** Data are means ± SEM.

published results with Mcam⁺ cells (Fig. 7f)[28]. Although engraftment potential substantially varied among the populations, qPCR analysis of sorted GFP⁺ cells from the host lungs showed that engrafted cells from all the groups upregulated *Cthrc1* and *Col1a1* compared to the untreated state (Fig. 7g, Supplementary Fig. 7b), suggesting that these resting populations all have the

potential to upregulate *Cthrc1* and *Col1a1* in response to local cues in the pro-fibrotic environment.

To explore molecular mechanisms that might be responsible for the emergence of cluster 8 cells, we isolated alveolar fibroblasts from untreated Col-GFP mice and treated the cells with growth factors or cytokines including TGF-β, TNF-α, and EGF in vitro.

We found that TGF-β stimulation induced several markers of cluster 8 cells, including *Col1a1*, *Cthrc1*, *Postn*, and *Tnc*, while TNF-α and EGF did not increase the expression of cluster 8 cell markers (Supplementary Fig. 7c). *Spp1* and *Fst* was not upregulated by TGF-β stimulation, suggesting that there are likely to be other factors in the fibrotic microenvironment involved in induction of the complete molecular phenotype of cluster 8 cells in vivo (Supplementary Fig. 7c).

**Purified *Cthrc1*+ fibroblasts show pathologic phenotype**. Next, we developed methods to purify *Cthrc1*+ pathologic fibroblasts from bleomycin-treated mice. Time course analysis revealed that the expressions of cluster 8 markers, such as *Cthrc1*, *Fst*, *Spp1* and *Col1a1*, peaked around day 10 after bleomycin treatment in Col-GFP+ cells (Supplementary Fig. 8a). In contrast, *Pdgfra* expression was downregulated after bleomycin treatment (Supplementary Fig. 8a). Flow cytometry of bleomycin-treated Col-GFP mice showed the emergence of a Sca1− Pdgfra− Col-GFP-high CD9-intermediate population not seen in untreated lungs (Fig. 8a). We purified this population, which we refer to as cluster 8 here, and were also able to purify each of the collagen-producing populations from the lungs of bleomycin-treated mice (Fig. 8a). qPCR analysis of purified cells showed that *Cthrc1* expression was highly enriched in cluster 8 (Fig. 8b). Consistent with our scRNA-seq data, *Col1a1* expression was the highest in cluster 8 and *Acta2* was enriched in Mcam+ cells (Fig. 8b). Of note, the markers for alveolar, adventitial, and peribronchial fibroblasts were similarly enriched by the same gating strategy used to purify the cells from untreated lungs (Supplementary Fig. 8b).

In fibrotic lungs, the cells that produce excess quantities of ECM are thought to arise in response to local injury and to migrate into injured areas where they form aggregates called fibroblastic foci that drive the fibrotic process[29]. We, therefore, sought to determine whether *Cthrc1*+ fibroblasts have increased the capacity to migrate in vitro and increased capacity to colonize the lungs of bleomycin-treated mice. We performed gap migration assays with cluster 8 cells, alveolar fibroblasts, and adventitial fibroblasts, which were purified from bleomycin-treated lungs on day 10 after treatment. Cluster 8 cells showed substantially higher migration capacity than alveolar or adventitial fibroblasts (Fig. 8c, d). We also performed the intratracheal transfer of cluster 8 cells, alveolar fibroblasts, or adventitial fibroblasts purified from bleomycin-treated lungs. Two days after the transfer, cluster 8 cells showed markedly higher engraftment potential than either alveolar or adventitial fibroblasts (Fig. 8e). Whole lung imaging of cleared host lungs also showed that transferred cluster 8 cells formed larger clusters in the host lungs than alveolar or adventitial fibroblasts (Fig. 8f). These results suggest that, in addition to expressing higher levels of genes encoding ECM proteins, *Cthrc1*+ fibroblasts have high migratory capacity and are better able to colonize fibrosing lungs. Taken together, our results suggest that *Cthrc1*+ fibroblasts are pulmonary fibrosis-associated fibroblasts with a highly activated phenotype.

**Discussion**
Our scRNA-seq results provide the first systematic atlas of the molecular characteristics and anatomic locations of collagen-producing cells in the adult lung. Our results confirm that smooth muscle cells and pericytes express low levels of collagen in both mice and humans, but that there are unique populations of cells that express higher levels of collagen and reside in distinct anatomic locations—the walls of conducting airways, surrounding the bronchovascular bundles, and embedded within the gas-exchanging alveolar region. We speculate that each population plays unique functional roles in organizing and maintaining the unique structures of each of these regions and in fibrotic responses to regional insults.

By comparing scRNA-seq results between normal and fibrotic lungs, we were also able to identify a unique population of cells that expressed the highest levels of collagens and other known components of the pathologic ECM, and were marked by high levels of expression of *Cthrc1* in both mice and human. These cells were present within clusters of collagen-producing cells in fibrotic regions, and displayed marked increase in their capacity to migrate and to colonize the fibrosing lung after intratracheal transfer, suggesting that they play important roles in driving pulmonary fibrosis.

In the absence of available lineage tracing mouse lines to definitively prove the origin of these cells, we used two different computational tools—RNA velocity and pseudotime trajectory analysis—which both suggested that these cells were most likely to have differentiated from a population of alveolar fibroblasts. Further support for this interpretation comes from our observations that levels of collagen expression progressively increased from populations of alveolar fibroblasts seen mainly in normal lungs, to populations of alveolar fibroblasts seen mainly in fibrotic lungs, to the *Cthrc1*-expressing cells. In parallel, expression of several markers shared by normal alveolar fibroblasts, including *Pdgfra*, *Tcf21*, and *Npnt* decreased progressively over this same continuum. Interestingly, both alveolar fibroblasts and adventitial fibroblasts purified from normal lungs and transferred back into the lungs of mice treated with intratracheal bleomycin were able to colonize the lungs of bleomycin-treated mice and markedly upregulated expression of *Cthrc1* after transfer, suggesting substantial plasticity of these populations in response to a fibrotic in vivo environment.

Myofibroblasts were first identified as contractile fibroblasts in granulation tissue[30]. After cutaneous injury, progenitor cells migrate into areas of injury and differentiate into myofibroblasts, which generate force and contract wounds[30]. An increase in the density of myofibroblasts in tissue is implicated in excessive collagen deposition in some disease settings[3,31]. In addition, adoptive transfer of purified fibroblasts in lung fibrosis showed upregulation of α-SMA together with ECM genes in transferred cells[28,32]. Our scRNA-seq data also show an increase in the number of *Acta2* expressing cells in *Cthrc1*+ fibroblasts compared to their presumed progenitor, alveolar fibroblasts. However, the highest level of α-SMA expression is observed in smooth muscle cells and pericytes, which display low levels of collagen expression. These findings suggest that α-SMA is unsuitable as a specific marker of pathologic ECM-producing cells in the lung.

We found that *Cthrc1* is a marker for pathologic fibroblasts in pulmonary fibrosis. *Cthrc1* is expressed in injured tissue and promotes cell migration[33,34]. Furthermore, invasive cancer cells express *CTHRC1* and high *CTHRC1* expression in tumors is associated with poor prognosis[35,36]. We found that *Cthrc1*+ fibroblasts from fibrotic lungs show high migration capacity and high potential to engraft when transferred into injured lungs. These findings are consistent with the previous suggestions that an invasive phenotype of fibroblasts promotes pulmonary fibrosis[37–40]. In humans with pulmonary fibrosis we and others have analyzed, *CTHRC1*+ fibroblasts emerged in pulmonary fibrosis associated with IPF and scleroderma, suggesting that there might be common mechanisms of fibrosis associated with *CTHRC1*+ fibroblasts in both diseases.

Lee et al. and Zepp et al. reported mesenchymal cells similar to the peribronchial fibroblasts we describe here, that they reported can differentiate into smooth muscle cells after airway injury[5,15]. Although our scRNA-seq also showed some shared genetic signature between peribronchial fibroblasts and smooth muscle cells,

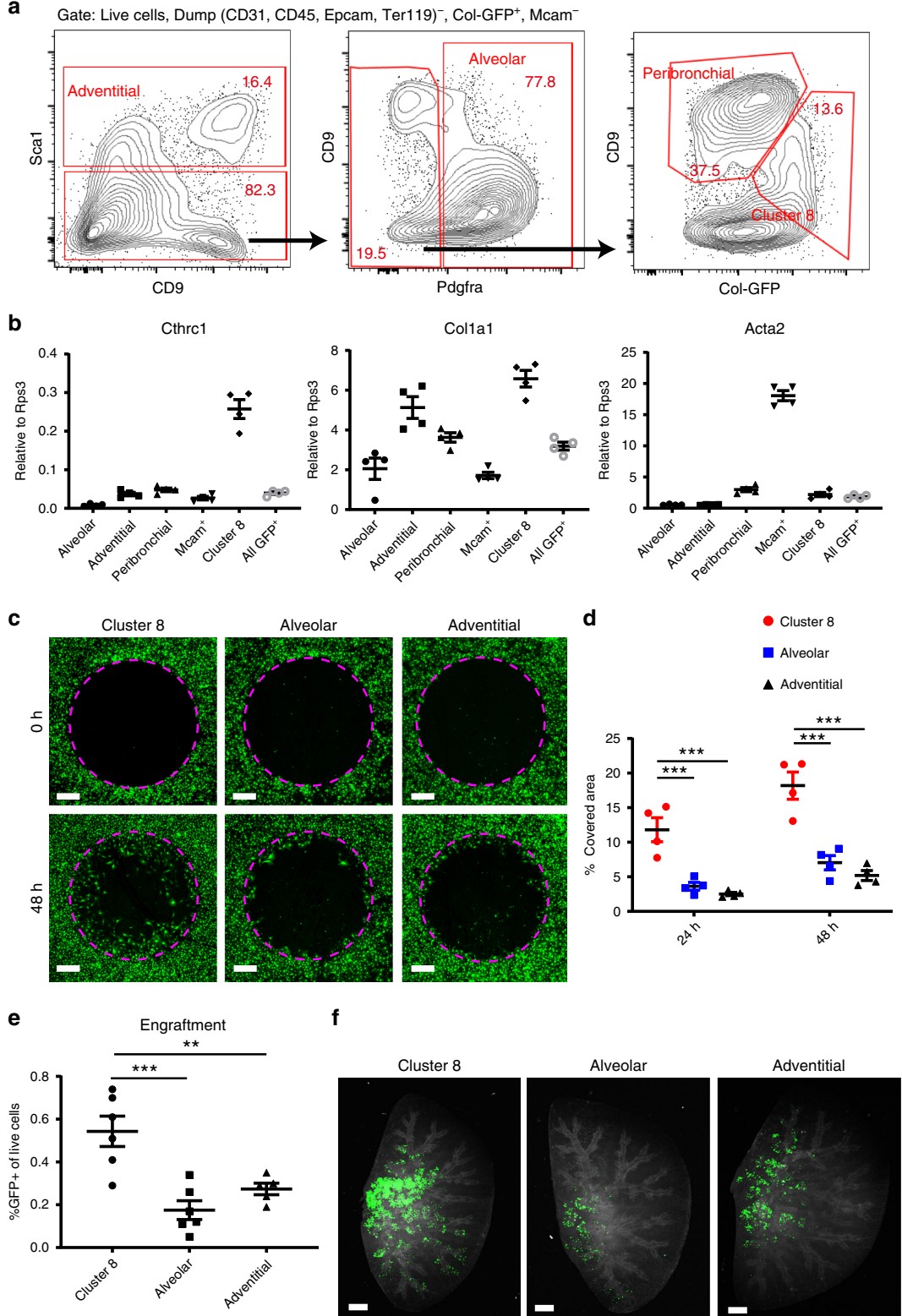

lineage tracing experiments will need to be performed to determine whether these cells can serve as smooth muscle progenitors in vivo. Our RNA velocity analysis did not show vectors towards smooth muscle cells from peribronchial fibroblasts but this could be because the rate of differentiation is too slow in normal or bleomycin-treated lungs.

Adventitial fibroblasts have been known to play important roles in tissue homeostasis and diseases[41]. Recent studies revealed

IL-33-expressing stromal cells support the development of group 2 innate lymphoid cells (ILC2) and form niches for ILC2 in peripheral tissue[11,42]. In addition to these $Il33^+$ adventitial fibroblasts, we revealed another kind of adventitial fibroblast characterized by $Adh7$ expression. The difference in physiological or pathological roles of these two adventitial fibroblasts is an intriguing question. Of note, we showed that adventitial fibroblasts have higher potential to engraft after transfer than alveolar

**Fig. 8 Purified Cthrc1+ pathologic fibroblasts showed high migration and invasion capacity. a** Gating strategy for purifying collagen-producing populations from bleomycin-treated lungs. **b** qPCR analysis of purified cells from bleomycin-treated lungs (day 10). $n = 4$ mice. **c** Representative images from gap migration assay with cells purified from bleomycin-treated lungs (day 10). Broken lines show initial cell-free zones. Col-GFP is shown in green. Scale bars, 500 μm. **d** Quantification of migration assay. $n = 4$ mice. ***$p < 0.001$, two-way analysis of variance followed by the Tukey–Kramer post-test. **e** The number of GFP+ cells from the host lungs, which received purified cells from bleomycin-treated lungs, were analyzed. $n = 5$ (adventitial) or 6 (cluster 8 and alveolar) mice. **$p < 0.01$, ***$p < 0.001$, one-way analysis of variance followed by the Tukey–Kramer post-test. **f** Whole lung imaging of host lungs. Col-GFP (donor cells) is shown in green. Autofluorescence in RFP channel is shown in white to visualize host lung cells. Images were maximum projection of z-stack images. Scale bars, 1 mm. **b–f** Data are representative from two experiments. **b, d, e** Data are means ± SEM. Source data are provided as a Source Data file.

---

fibroblasts. Taken together with a study showing Sca1+ stromal cells have stem cell-like phenotypes and are multipotent[43], adventitial fibroblasts might also serve as progenitors for pathologic fibroblasts, perhaps in response to insults arising in proximal regions of the lung or insults arising from the lung vasculature.

In conclusion, we have built an atlas of collagen-producing cells and identified several molecularly distinct cell types that occupy unique anatomic locations. We have also identified a novel population of cells unique to the fibrotic lung that produce the highest levels of collagens and other ECM proteins, and show the highest capacity to migrate and to colonize the fibrosing lung. The development of tools to mark, ablate and modify gene expression in each of these subtypes should provide important insights into the lineage relationships, homeostatic and pathologic roles of each of these cell types.

## Methods

**Mice and bleomycin treatment.** Col-GFP mice were obtained from Dr. David Brenner at University of California, San Diego, CA, and maintained on the C57BL/6J background[8]. Shh-Cre (Stock No. 005622) and Rosa26-lox-stop-lox-tdTomato (Stock No. 007914) mice were obtained from the Jackson Laboratory. Although the Cre protein is fused with GFP in Shh-Cre mice, the expression of Shh is restricted to epithelial cells and Col-GFP+ fibroblasts can be identified as GFP single-positive cells while epithelial cells are all tdTomato+ in Col-GFP/Shh-Cre/Rosa26-lox-stop-lox-tdTomato mice. We obtained C57BL/6J mice from the Jackson Laboratory (stock no. 000664) as host mice for adoptive transfer experiments. Sex-matched mice between the ages of 8 and 12 weeks old were used for the experiments. Female mice were used for scRNA-seq. Both male and female mice were used in the other experiments. For fibrosis induction, mice were treated with 3 U/kg bleomycin in 70 μl saline by oropharyngeal aspiration, except host mice for transfer experiments, which were treated with 2 U/kg bleomycin. Mice were maintained in the UCSF specific pathogen-free animal facility in accordance with guidelines established by the Institutional Animal Care and Use Committee and Laboratory Animal Resource Center. All animal experiments were in accordance with protocols approved by the University of California, San Francisco Institutional Animal Care and Use Committee.

**Tissue dissociation.** Mouse lungs were harvested after perfusion through the right ventricle with 5 ml PBS. For human lungs, representative pieces were collected from whole lungs and cut into $5 \times 5 \times 5$ mm pieces. Approximately 1 g tissue was randomly chosen from the pieces for digestion. After mincing with scissors, the tissue was suspended in protease solution [0.25% Collagenase A (Millipore Sigma), 1 U/ml Dispase II (Millipore Sigma), 2000 U/ml Dnase I (Millipore Sigma) in RPMI (Millipore Sigma) supplemented with 10 mM HEPES]. The suspension was incubated at 37 °C for 60 min with trituration by micropipette every 20 min. Then the cells were passed through 100 μm cell strainer (BD Biosciences), washed with PBS, and suspended in PBS with 0.5% bovine serum albumin (BSA) (Fisher BioReagents).

**Flow cytometry.** For scRNA-seq in mice, GFP+ and GFP− cells were sorted after gating singlet, live (DAPI−), and Ter119−. For scRNA-seq in human, lineage negative (CD31, CD45, EPCAM, CD235a)− or all lung cells (CD235a−) were sorted after gating singlet and live cells (DAPI−). Donor cells for transfer experiments were sorted following magnetic negative selection for CD31, CD45, Epcam, and Ter119 by biotin-labeled antibodies and Dynabeads MyOne Streptavidin T1 (Thermo Fisher Scientific). The antibodies used were as follows: anti-CD9 (clone MZ3, PE; BioLegend), anti-CD31 (clone 390, biotin; BioLegend), anti-CD45 (30F-11, biotin; BioLegend), anti-Pdgfra (clone APA5, APC; BioLegend), anti-Mcam (clone ME-9F1, PerCP/Cy5.5; BioLegend), anti-Epcam (clone G8.8, biotin; BioLegend), anti-Sca1 (clone D7, PE/Cy7; BioLegend), streptavidin-APC/Cy7

(BioLegend), anti-hCD31 (clone WM59, PE/Cy7; BioLegend), anti-hCD45 (clone HI30, APC; BioLegend), anti-hEPCAM (clone 9C4, FITC; BioLegend), anti-hCD235a (clone HI264, APC/Cy7; BioLegend). Sorting was performed using FACS Aria III (BD Biosciences). Flow cytometry data were analyzed using the FlowJo v10.

**Human lung tissues.** The studies described in this paper were conducted according to the principles of the Declaration of Helsinki. Written informed consent was obtained from all subjects, and the study was approved by the University of California, San Francisco Institutional Review Board. Fibrotic lung tissues were obtained at the time of lung transplantation from patients with a diagnosis of usual interstitial pneumonia or scleroderma. Normal lung tissues were obtained from lungs rejected for transplantation by the Northern California Transplant Donor Network.

**scRNA-seq library preparation and sequencing.** We collected lung cells from two untreated and two bleomycin-treated (day 14) mice. After gating live cells and Ter119− cells, GFP+ and GFP− cells were sorted and each sample were processed separately. For human samples, lineage negative (CD31, CD45, CD235a, EPCAM)− cells or all lung cells (CD235a−) were sorted. Approximately $1 \times 10^5$ cells were sorted for each sample. The sorted cells were re-suspended in PBS with 0.05% BSA and ~$1 \times 10^4$ cells were loaded onto the Chromium controller (10x Genomics). Chromium Single Cell 3' v2 reagents were used for library preparation according to the manufacturer's protocol. For human samples, the libraries were stored at −20 °C and 2–3 batches were pooled for sequencing. The libraries were sequenced on Illumina HiSeq 4000 for mouse samples or NovaSeq 6000 for human samples.

**Sequencing data processing for mouse data.** Sequencing data were aligned to mouse genome mm10 with Cell Ranger version 2.0 (10x Genomics). The data were processed using the Seurat R package version 2.3.4[44]. We excluded cells with fewer than 250 detected genes or larger than 10% percent mitochondria genes. Samples from untreated and bleomycin-treated lungs were aggregated with CCA subspace alignment[9]. For *Col1a1*+ cell analysis, we aggregated GFP+ samples with CCA subspace alignment, and clusters which expressed lineage markers for hematopoietic, endothelial, epithelial, and mesothelial cells were removed. *Col1a1* expression was not consistently detectable among each of these removed clusters except the small mesothelial cluster. Variable genes for dimensionality reduction were selected using *FindVariableGenes* function of Seurat. We performed graph-based clustering of the CCA reduced data using *FindClusters* function of Seurat (resolution = 0.6, dims.use = 1:20). Visualization of the clusters on a 2D map was performed with uniform manifold approximation and projection (UMAP) (*RunUMAP* function of Seurat, dims.use = 1:20). Violin plots and UMAP plots overlaid with gene expression level were generated using Seurat. Differentially expressed genes of each cluster were identified using *FindAllMarkers* function of Seurat.

**RNA velocity and pseudotime trajectory analysis.** We used the velocyto R package v0.6 to estimate cell velocities from their spliced and unspliced mRNA content[18]. We generated annotated spliced and unspliced reads from the 10x BAM files via the dropEst pipeline, before calculating gene-relative velocity using kNN pooling with k = 25, fitting gene offsets using spanning reads and a gamma fit on the top/bottom 2% expression quantiles, and determining slope gamma with the entire range of cellular expression. We visualized aggregate velocity fields (using Gaussian smoothing on a regular grid) on the UMAP visualizations generated previously in Seurat. To generate cellular pseudotemporal trajectories we used the monocle R package v2.6.4[19]. We ordered cells in an unsupervised manner (using genes with mean expression >1 and empirical dispersion ≥2*dispersion model estimate) and scaled the resulting pseudotime values between 0 and 1.

**Sequencing data processing for human data.** Sequencing data were aligned to human genome GRCh38 with Cell Ranger version 2.0 or 3.0. The data were first processed using the scran R package v1.10[22]. We excluded cells which were five median absolute deviation distant from the median value of library size, number of detected genes, or mitochondrial gene proportion. Variable genes were identified as genes with the largest biological components across all the samples following

decomposition of the variance in expression for each gene into biological and technical components as described previously[45]. All samples were aggregated with MNN-batch correction[22]. Aggregated data and MNN dimensionality reduction were imported into Seurat. We performed graph-based clustering of the MNN reduced data using *FindClusters* function of Seurat (resolution = 0.3, dims.use = 1:19). For lineage sample analysis, clusters which expressed epithelial and endothelial markers were removed. Visualization of the clusters on a 2D map was performed with UMAP (*RunUMAP* function of Seurat, dims.use = 1:19). Violin plots and UMAP plots overlaid with gene expression level were generated using Seurat. Differentially expressed genes of each cluster were identified using *FindAllMarkers* function of Seurat.

**Sequencing data processing for the large data set.** scRNA-seq data of IPF and normal human lungs from the IPF Cell Atlas preprint[23]. Following unsupervised clustering, stromal populations were identified on the basis of positive PDGFRB expression, resulting in a total of 6607 cells: 1144 control cells from 26 subjects and 5463 cells from 32 IPF patients. Gene expression was normalized by first scaling absolute UMI values to relative values of per 10,000 transcripts, followed by natural log transformation with a pseudocount of 1 (i.e., log((TPM/100)+1). Amongst the PDGFRB stromal cells, expressing was normalized with SCTransform 0.2.0 for the top 10,000 defined variable genes to regress out signals associated with the fraction of mitochondrial transcripts and the overall number of unique molecular identifiers (UMIs), using seed:7. Variable features were selected with Seurat 3.1.0 using the "mean variance plot" method, with a mean expression cutoff range between 0.15 and 4, and dispersion cutoff between 0.8 and 7. Amongst selected variable genes, only genes normalized by SCTransform were selected for PCA analysis under the default Seurat parameters. The top 6 dimensions were used for UMAP embedding with a neighborhood size of 20, a minimum distance of 0.1, for 1000 epochs. UMAPs featuring gene expression show the log((TPM/100)+1) values of expression per cell.

**Sequencing data processing for publicly available data sets.** Gene expression matrices of Peyser et al. (GSE129605)[20] and Morse et al. (GSE128033)[25] were processed using the Seurat R package version 2.3.4. Variable genes for dimensionality reduction were selected using *FindVariableGenes* function of Seurat. The samples were aggregated with *RunMultiCCA* function of Seurat. After performing graph-based clustering of the CCA reduced data using *FindClusters* function of Seurat, *Col1a1*-expressing clusters were picked up for further analysis. UMAP plots overlaid with gene expression level were generated using Seurat.

**PLISH.** PLISH was performed as described previously with some modifications[10]. Lungs were fixed with 4% paraformaldehyde (PFA) for 6 h on ice and dehydrated with 30% sucrose overnight at 4 °C. Sixteen μm sections were made from optimal cutting temperature compound (Sakura Finetek)-embedded frozen blocks. Target retrieval was performed by treating the sections with 10 mM citrate buffer containing 0.05% lithium dodecyl sulfate at 100 °C for 5 min and with 0.25 mg/ml pepsin in 0.1 M HCl at 37 °C for 5 min. The sections were incubated with H-probes in H-probe buffer at 37 °C for 2 h, followed by incubation with phosphorylated common bridge and circle oligonucleotides at 37 °C for 1 h. After incubation in ligation buffer at 37 °C for 2 h, the sections were incubated in RCA buffer at 37 °C overnight. Then the sections were incubated in labeling buffer containing Cy3 and Cy5-labeled oligonucleotide at 37 °C for 1 h. After washing with PBS with 0.05% tween 20, the sections were stained with chicken anti-GFP antibody (ab13970, Abcam) at room temperature for 1 h, followed by incubation with donkey anti-chicken alexa 488 secondary antibody (Jackson Immuno Research) and DAPI (Thermo Fisher) at room temperature for 1 h. The sections were mounted with ProLong Glass antifade Mountant (ThermoFisher). Images were taken using Zeiss 780 confocal microscope. Oligonucleotide sequences used are listed in Supplementary Table 1.

**Immunohistochemistry (IHC) and ISH staining for human samples.** Chromogenic IHC and in situ hybridization (ISH) staining for human samples were performed on an automated Bond RX immunostainer (Leica) as described below. 4 μm microtome sections of formalin-fixed, paraffin-embedded (FFPE) healthy and diseased IPF lung were collected on slides for IHC and ISH staining. For IHC staining, heat-induced epitope retrieval was performed by treating the sections with citrate buffer at 90 °C for 30 min. Sections were incubated with rabbit anti-CTHRC1 antibody (ab85739, Abcam) or anti-α-SMA antibody (ab5694, Abcam) at ambient temperature for 30 min, followed by Bond Polymer Refine Red Detection (DS9390, Leica). ISH staining was performed following Advanced Cell Diagnostic's (ACD) standard Bond RX automated ISH protocol using RNAscope® 2.5 LS Reagent Kit-RED (Advanced Cell Diagnostics) and Bond Polymer Refine Red Detection (DS9390, Leica). RNAscope® LS 2.5 Probes against human CTHRC1 (413338, ACD) were ordered from ACD with hybridization optimized for human skin (EDTA for 15 min at 88 °C). Stained slides were digitized for image analysis using a Perkin Elmer P250 whole slide digital pathology slide scanner with 20×(IHC) or 40×(ISH) objective and extended focus scanning parameters.

**Imaging cleared lung tissues.** Untreated lungs of Col-GFP mice were fixed with 4% PFA overnight at 4 °C and then inflated with low-melting point agarose. 100 μm sections were made using a vibratome VT1000S (Leica). The sections were cleared using a CUBIC method as described previously[46]. After delipidation with Reagent-1A (10 wt% triton, 5 wt% NNNN-tetrakis (2-HP) ethylenediamine, 10 wt% urea, 25 mM NaCl), the sections were stained with anti-α-SMA-alexa 647 (R&D), or anti-collagen 4 (LSL) followed by donkey anti-rabbit IgG-alexa 647 (Thermo Fisher). The sections were then treated with refractive index-matching reagent (Reagent-2; 25 wt% urea, 50 wt% sucrose, 10 wt% triethanolamine) and imaged using W1 spinning disk confocal microscope (Nikon). Images were processed using Image J version 1.52i. 3D-reconstruction of z-stack image was performed using Icy version 2.0. For whole lung imaging after transfer, 4% PFA-fixed lungs were cleared with Reagent-1A and treated with Reagent-2, followed by imaging for GFP signal using Nikon AZ100 microscope configured as light sheet microscopy. Autofluorescence signal in RFP channel was used to visualize the lung structure. Maximum projection images were generated using Image J.

**Quantitative real-time PCR analysis.** Approximately 2000 cells were directly sorted into TRIzol reagent (Thermo Fisher), and RNA was isolated according to the manufacturer's protocol. The RNA was reverse-transcribed using a Super Script IV VILO Master Mix with ezDNase Enzyme kit (Thermo Fisher). Quantitative Real-Time PCR was performed using 2x SYBR Green qPCR Master Mix (Bimake) with an ABI 7900HT (Applied Biosystems). Primer sequences are listed in Supplementary Table 2.

**Primary alveolar fibroblast in vitro stimulation.** Alveolar fibroblasts were sorted from untreated Col-GFP mice. $1 \times 10^5$ cells were seeded into 48 well plates and cultured in DMEM (Corning) with 2% fetal bovine serum (FBS) (Gibco) and 1% penicillin–streptomycin (Gibco) for 24 h. Then medium was changed to serum-free DMEM with 1% penicillin/streptomycin for 24 h. After the serum starvation, medium was changed to serum-free DMEM with 1% penicillin/streptomycin, containing 1 ng/ml TGF-b (R&D), 10 ng/ml TNF-a (R&D), or 10 ng/ml EGF (Thermo Fisher). After 24 h stimulation, cells were lysed by directly adding 500 μl Trizol into the wells. Cell culture was performed under standard conditions (37 °C, 5% $CO_2$).

**Intratracheal adoptive transfer.** Donor cells were collected from untreated or bleomycin-treated (day 10) Col-GFP mice. Fibroblast subpopulations were sorted into PBS with 0.5% BSA. After resuspending the sorted cells in smaller volume of PBS with 0.5% BSA, live cell number was counted and cell concentrations were adjusted so that each sample had identical cell concentration. 70 μl of cell suspension was instilled into bleomycin-treated wild type mice (day 7) by oropharyngeal aspiration. $1–3 \times 10^5$ cells were transferred per mouse.

**Migration assay.** Fibroblast subpopulations were sorted into PBS with 0.5% BSA. $1 \times 10^5$ cells were resuspended in 100 μl DMEM (Millipore Sigma) with 2% fetal bovine serum (FBS) and dispensed into each well of Oris cell migration assay kit (fibronectin-coated) (Platypus Technologies). After 24 h incubation in a $CO_2$ incubator, the stoppers were removed and the wells were refilled with 100 μl DMEM with 2% FBS after washing with 100 μl PBS. The plate was incubated in a $CO_2$ incubator and GFP+ cells were imaged after 24 and 48 hr using Zeiss spinning disk confocal microscopy. GFP+ area in the central circle were quantified using Image J.

**Data analysis.** scRNA-seq data analysis was performed in R version 3.5.1. Statistical tests were performed in GraphPad Prism version 8.1.2.

**Reporting summary.** Further information on research design is available in the Nature Research Reporting Summary linked to this article.

## Data availability

The scRNA-seq data generated in this study are deposited in Gene Expression Omnibus (GEO) (GSE132771). The mesenchymal subset of large scRNA-seq data set of 29 normal and 32 IPF lungs[23] is deposited in GEO (GSE147066). An interactive online tool for the data set is available at www.IPFCellAtlas.com. The source data underlying Figs. 7d, f, g, 8b, and e, and Supplementary Figs. 7a–c, 8a and b are provided as a Source Data file.

## Code availability

The codes used in scRNA-seq analysis are available from the authors upon reasonable requests.

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

## Acknowledgements

T.T. was supported by Japan Society for the Promotion of Science (JSPS Overseas Research Fellowship), the Uehara Memorial Foundation, Mochida Memorial Foundation for Medical and Pharmaceutical Research, and the Frontiers in Medical Research Fellowship from the California Foundation for Molecular Biology. K.S. was supported by an F32 award from the NHLBI. N.C.H. was supported by a Wellcome Trust Senior Research Fellowship in Clinical Science (ref. 103749). We thank Dr. Walter Eckalbar at UCSF for support with computational analysis. We thank the Nina Ireland Program for Lung Health for support for sourcing human tissues. This work was supported by HL123423, HL108794, a grant from AbbVie (D.S.), R01HL127349, U01HL122626 (N.K.), NHLBI P01 HL114501, support from the Pulmonary Fibrosis Fund (I.O.R), an unrestricted gift from Three Lake Partners (I.O.R. and N.K.). We thank UCSF core facilities: Laboratory for Cell Analysis for FACS and microscopes, Nikon Imaging Center for microscopes, Gladstone Genomics core and Institute of Human Genetics for scRNA-seq.

## Author contributions

T.T., K.S., and D.S. conceived the study. T.T., K.S., and J.B.W. performed the experiments. T.T., K.S., and J.W.K. performed the bioinformatics analysis. L.A.H. and N.C.H. supervised select experiments and analyses. M.A.M. and P.J.W. procured human samples. T.S.A., J.C.S., S.P., I.O.R., and N.K. generated and analyzed the scRNA-seq data set of 29 normal and 32 IPF lungs. T.T. and D.S. wrote the paper with input from co-authors. D.S. supervised the study.

## Competing interests

D.S. is a founder of Pliant Therapeutics and has received research funding from Abbvie, Pfizer and Pliant Therapeutics. He serves on the Scientific Review Board for Genentech, xCella Biosciences and Optikira. M.A.M. has received research funding from Bayer Therapeutics and is a consultant for Cerus Therapeutics and Gen1e Life Sciences. T.S.A., J.C.S., S.P., I.O.R., and N.K. are inventors on a provisional patent application (62/849,644), Yale University and Brigham and Women's Hospital, Inc. that covers methods related to IPF associated cell subsets. N.K. served as a consultant to Biogen Idec, Boehringer Ingelheim, Third Rock, Pliant, Samumed, NuMedii, Indaloo, Theravance, LifeMax, Three Lake Partners, Optikira over the last 3 years and received non-financial support from MiRagen.
