## [Peer Review File · Nature Communications]

Reviewers' comments:

Reviewer #1 (Remarks to the Author):

Thank you for asking me to review this paper by Tsukui et al titled "collagen producing lung cell atlas identifies multiple subsets with distinct localization and relevance to fibrosis". The authors initially utilise single cell RNAseq data in the murine bleomycin-induced lung fibrosis and in lung samples from IPF, scleroderma and control subjects to identify unique clusters of collagen producing cells in the lung. They further use specific imaging techniques to identify the geographic localization of these different clusters in the lung and undertake adoptive transfer experiments to assess the role of these cells in fibrogenesis. The paper presents a large body of work which is logically and clearly presented. The RNAseq data is supported by the in vivo imaging and adoptive transfer experiments supporting the validity of the clusters identified by single cell analysis. Overall, the results are of considerable interest and increase our understanding of the different cell types involved in the pathogenesis of fibrotic lung disease. Confidence in the validity of the results is enhanced by the in vitro and in vivo validation work performed. The complexity of the bioinformatic approaches used to undertake the analyses is beyond my level of expertise and I would suggest getting a review by a bioinformatician who is comfortable in the analysis and interpretation of single cell RNAseq data.

As noted, I feel the flow of experiments performed is logical and well presented. My only (minor) reservation is that the number of human samples used is small and these come from two similar, but ultimately distinct, diseases – IPF and scleroderma. A number of single cell RNAseq studies have recently been published and have provided publically available datasets (for instance, Reyfman et al AJRCCM 2019). The authors findings would be considerably strengthened if they could "validate" their findings in an external dataset.

Reviewer #2 (Remarks to the Author):

The present study sets out to develop a collagen-producing lung cell atlas using state-of-the-art single cell sequencing and informatics approaches. The mouse model data are clear, compelling and of high quality. Novel Col+ subpopulations are identified and immunolocalized in the mouse lung, and previously described subpopulations are confirmed. Only a few minor technical issue (detailed below) require attention. Where the study falls short in my view is on the human relevance side of the ledger. The subpopulations identified from human samples are not functionally characterized nor are they immunolocalized in the lungs from which they were derived. In addition, the rationale for studying cells from 2 different human lung disorders (3 IPF patients and 2 scleroderma patients) is not provided. Although both are fibrotic lung disorders, the patterns of fibrosis differ and the authors do not account for, nor acknowledge this. The manuscript would be stronger and of much greater interest to the broad scientific community with 3 important additions:

1. Functional data for the human cluster 8 cells (invasion, adoptive transfer);
2. Immunolocalization of the human subpopulations in the parent lung (e.g., do they localize to areas of active ECM synthesis?);
3. A large enough sample size of at least one of the 2 human lung diseases studied to parse out individual effects from disease class effects.

Specific comments

Figs 2 – I assume these are untreated lungs. Please state this in the Fig legend

Fig 4 – Since these are fibrotic lungs, a serial H and E section from the sequence shown in C is needed for morphological orientation – although a video as for the untreated lungs would be nice.

Fig 7 – Adding a measurement of lung collagen would be helpful to assess the ability of cluster 8 cells to produce collagen after engraftment

Reviewer #3 (Remarks to the Author):

In this manuscript, the authors described the scRNA-seq analysis using the mouse model of bleomycin-induced lung fibrosis. They particularly focused on Col1a1+ mesenchymal cells and classified them into 12 clusters. They further conducted the PLISH analysis, which is an RNA in situ hybridization method, for detecting the physical localization of the cell population of each cluster. By the series of the analyses, the authors concluded that Cthrc1+ cells (Cluster 8), which presumably descendent from alveolar fibroblasts, are pathogenic in bleomycin-induced lung fibrosis. They also conducted the similar scRNA-seq analysis and found that human CTHRC1+ cells are specific to fibrotic lungs in patients with idiopathic pulmonary fibrosis (IPF) and scleroderma. They further characterized Cthrc1+ cells in migration assays in vitro and intratracheal transfer experiments in vivo, demonstrating that Cthrc1+ cells are highly migratory and invasive after intra tracheal transfer to bleomycin-induced lung fibrosis mice.

Overall, I admit this is a high quality paper. However, I'm afraid I have to suggest that the idea of Cthrc1 as a biomarker somewhat lacks novelty as in the light of the previous publications of scRNA-seq using the same model. Moreover, the results from the previous papers seem not always consistent with those described in the present paper. Therefore, I consider further careful extensive analyses should be needed.

General:

At least two papers, in my knowledge, have already reported the scRNA-seq of the bleomycin-induced lung fibrosis mice (Xie, et al. Cell Rep, 2018; Peyser, et al. Am J Respir Cell Mol Biol, 2019). These two studies have also reported the heterogeneity of the activated fibroblasts. Particularly, Peyser et al. have described that Cthrc1 should be elevated in the activated fibroblasts as well as other biomarkers. However, they concluded that pathogenic fibroblasts could not be defined as a uniquely population. On the other hand, the present study identified that a unique population of Cthrc1+ cells (Cluster 8) are more pathogenic than the other fibroblasts. To judge which claims should be rationalized, I'm not convinced that the evidence presented in this paper should be sufficiently strong, considering the presented experiments are mostly observatory (Figures 1-5). The migration assays in vitro and intratracheal transfer experiments should be the key. However, they are not conclusive how their Cthrc1+ cells affect bleomycin-induced lung fibrosis and human lung fibrosis.

Generally, I'd like to suggest that the present study should put a more focus on further detailed study of the human fibroblasts of the lung fibrosis patients. I'd be particularly interested in seeing further proof-of-concept study, such as contemplating a therapeutic strategy which may target Cthrc1+ cells (Cluster 8).

Major comments:

1. The authors should have compared the results with previous studies in more details. They attempted to explore the role of Cthrc1+ cells but I'd like to suggest that the present data is insufficient to argue whether Cthrc1 suppress or exacerbate the collagen accumulation in bleomycin-induced lung fibrosis and human lung fibrosis. Further enriched data supporting the consistency between mouse and human CTHRC1+ fibroblasts should be also needed.

2. I'm concerned that the process of the scRNA-seq data should be somewhat arbitrary. For example, the authors employed the t-SNE method for initial clustering and the UMAP method for the second clustering. For the mouse cells, they used the CCA method for the batch-effect correction, while, for the humans, the MNN method was used, which have different characteristics. Especially, please carefully check the results between mice and humans should be compared,

nevertheless.

3. Molecular mechanisms underlying the observed phenotypic appearance of the Cthrc1 cells should be further explored. Otherwise, it would be difficult to identify these cells as the truly important cellular populations in the human fibrosis.

Minor comments:

4. Figure 1: For the scRNA-seq, I'm not convinced that the identified Col1a1+ cells should represent the entire population of collagen producing fibroblasts. The authors stated "we sorted GFP- cells to compare gene expression patterns in the cells responsible for collagen production to pattern seen in other cell types," I could not find the data. Please further explain the rationale in analyzing the Col1a1+ cells over the other pan-fibroblast markers such as, CD90 and Vimentin, in order to avoid the selection biases.

5. Generally, the number of cells are described as "approximately". Perhaps in Supplementary File, please provide the exact number how many cells belong to each of the clusters. If the authors employ different methods or the parameters for each dataset, the robustness of the results should be evaluated in a quantitative manner.

6. pp 9 and Figure 4: The authors suggested that the Col1a1+ cells may be derived from alveolar fibroblasts based on the results of RNA velocity and pseudotime trajectory analyses. I wonder if the authors could actually differentiate the Col1a1+ cells into the Cthrc1+ cells, for example, by TGFB1 or any other fibrosis-related cytokine treatments. TGFb stimulation can sometimes convert the lipofibroblasts to myofibroblasts (Agha, et al. Cell Stem Cell, 2017).

7. The data of human CTHRC1+ fibroblasts are just an observation study, which should become more convincing to ensure the consistency between the mouse Cthrc1+ cells in Cluster 8 and the human CTHRC1+ fibroblasts. For example, a similar immunostaining analysis or PLISH analysis for CTHRC1 in the human specimen of IPF should be needed. Positional relationship between the human CTHRC1+ cells and the myofibroblasts is particularly interesting.

8. Figure 6B: Violin plots of each marker, as shown in Figures 2A and 4A, would be helpful to understand the results in a more quantitative manner.

9. Figure 6C: The data clearly represents the presence of two populations by the gating of "Alveolar"; one is CD9high and the other is CD9low. What is the CD9high PDGFR+ population? Similarly, in the last panel of Figures 6C and 7C, there is a CD9 negative population. What is that population?

10. It is preferable to include the migratory assay for the human CTHRC1+ fibroblasts as well

11. Figure 6E: How would the Cluster 8 cells affect the bleomycin-induced lung fibrosis? Would they suppress or exacerbate the fibrosis? The role of Cthrc1 in the Cluster 8 cells should be more clearly addressed.

12. Figures 7C, D, E and F: What are the indications obtained from the results of the peribronchial fibroblasts and Mcam+ cells.

Miscellaneous comments:

13. pp7, line 146: Please correct "Figure 2D" to "Figure 3D"

14. pp9, line 186-192: The authors address that cluster 0 were differentiated into clusters 1 and 2, then cluster 2 differentiated into cluster 8. Considering that the RNA velocity data (Figure 4D), cluster 1 seems to be converted to cluster 0 and 2, then cluster 2 turned into cluster 8. Is that right?

Response to reviewers' comments:

Reviewer #1 (Remarks to the Author):

Thank you for asking me to review this paper by Tsukui et al titled “collagen producing lung cell atlas identifies multiple subsets with distinct localization and relevance to fibrosis”. The authors initially utilise single cell RNAseq data in the murine bleomycin-induced lung fibrosis and in lung samples from IPF, scleroderma and control subjects to identify unique clusters of collagen producing cells in the lung. They further use specific imaging techniques to identify the geographic localization of these different clusters in the lung and undertake adoptive transfer experiments to assess the role of these cells in fibrogenesis. The paper presents a large body of work which is logically and clearly presented. The RNAseq data is supported by the in vivo imaging and adoptive transfer experiments supporting the validity of the clusters identified by single cell analysis. Overall, the results are of considerable interest and increase our understanding of the different cell types involved in the pathogenesis of fibrotic lung disease. Confidence in the validity of the results is enhanced by the in vitro and in vivo validation work performed. The complexity of the bioinformatic approaches used to undertake the analyses is beyond my level of expertise and I would suggest getting a review by a bioinformatician who is comfortable in the analysis and interpretation of single cell RNAseq data.

As noted, I feel the flow of experiments performed is logical and well presented. My only (minor) reservation is that the number of human samples used is small and these come from two similar, but ultimately distinct, diseases – IPF and scleroderma. A number of single cell RNAseq studies have recently been published and have provided publically available datasets (for instance, Reyfman et al AJRCCM 2019). The authors findings would be considerably strengthened if they could “validate” their findings in an external dataset.

We thank reviewer #1 for suggesting that we analyze publicly available datasets to validate our conclusions about human cells, which were based on a small number of human samples. Reyfman et al AJRCCM 2019 did not sequence many fibroblasts, possibly because of the dissociation method they used. Instead, we have analyzed another recently published data set (Morse et al. Eur Respir J 2019. ref 25). We were able to identify a subset of COL1A1⁺ cells from IPF lungs that co-expressed CTHRC1 and other ECM-related genes, consistent with our findings. This analysis is shown in Supplementary Fig. 6b. Moreover, we collaborated with a group from Yale that has performed scRNA-seq in 29 normal and 32 IPF lungs (ref. 23), and together we re-analyzed their data clustering all COL1A1-expressing cells, as we have done for our initial data. This analysis confirmed the presence of clusters with the same gene expression patterns we identified (pp 12 line 240). We also cited another recently published paper, which showed similar heterogeneity of lung fibroblasts from 5 normal and 8 scleroderma patients (pp 13 line 246. ref. 24)

Reviewer #2 (Remarks to the Author):

The present study sets out to develop a collagen-producing lung cell atlas using state-of-the-art single cell sequencing and informatics approaches. The mouse model data are clear, compelling and of high quality. Novel Col⁺ subpopulations are identified and immunolocalized in the mouse lung, and previously described subpopulations are confirmed. Only a few minor technical issue (detailed below) require attention. Where

the study falls short in my view is on the human relevance side of the ledger. The subpopulations identified from human samples are not functionally characterized nor are they immunolocalized in the lungs from which they were derived. In addition, the rationale for studying cells from 2 different human lung disorders (3 IPF patients and 2 scleroderma patients) is not provided. Although both are fibrotic lung disorders, the patterns of fibrosis differ and the authors do not account for, nor acknowledge this.

We thank reviewer #2 for bringing our attention to not providing the rationale for collecting samples from IPF and scleroderma patients. We collected samples from two diseases to explore whether the CTHRC1⁺ population of apparently fibrogenic fibroblasts we identified from the mouse bleomycin model might be a common feature in human diseases characterized by pulmonary fibrosis. Indeed, our results suggest that the emergence of CTHRC1⁺ pathologic fibroblasts is common between the mouse model and both human diseases. We have added the rationale to pp 11, line 215. As noted above, in response to comments from the editor and reviewer 1, we have now expanded our analysis to a much larger group of patients with IPF by analyzing data made available from our collaborators from single cell sequencing from 29 normal subjects and 32 patients with IPF, and confirmed the findings that we found in our initial small number of patients.

The manuscript would be stronger and of much greater interest to the broad scientific community with 3 important additions:

1. Functional data for the human cluster 8 cells (invasion, adoptive transfer);

While we agree that it will also be important to characterize in vitro behavior of these human cells, and are working hard on accomplishing this goal, we have not yet perfected methods to purify these cells from human explants and fear that delaying our paper until we accomplish this important goal might negatively impact the timeliness of our findings.

2. Immunolocalization of the human subpopulations in the parent lung (e.g., do they localize to areas of active ECM synthesis?);

We performed in situ hybridization and antibody staining for CTHRC1 in IPF lung sections. CTHRC1⁺ cells were mainly localized within fibroblastic foci, which are unique pathological feature of IPF and known as active sites of collagen production. These results strengthen our conclusion that CTHRC1⁺ fibroblasts are pathologic fibroblasts at the site of collagen production. We have added these panels to Fig. 5h.

3. A large enough sample size of at least one of the 2 human lung diseases studied to parse out individual effects from disease class effects.

As noted, above, we have added analysis of new publicly available data sets and analysis of data on a much larger group of normal patients and patients with IPF from our collaborators to validate our findings as reviewer #1 suggested (pp 12 line 240. Supplementary Fig. 6a-d). We also now cite a recently-published paper, in which authors performed scRNA-seq on cells from 5 normal and 8 scleroderma lungs and found emergence of a similar population of CTHRC1⁺ cells (pp 13, line 246, ref. 24.)

Specific comments

Figs 2 – I assume these are untreated lungs. Please state this in the Fig legend

We have added “untreated” in the figure 2 legend tile.

Fig 4 – Since these are fibrotic lungs, a serial H and E section from the sequence shown in C is needed for morphological orientation – although a video as for the untreated lungs would be nice.

We have added an image of H&E staining from the serial section (Fig. 4c)

Fig 7 – Adding a measurement of lung collagen would be helpful to assess the ability of cluster 8 cells to produce collagen after engraftment

Because our transfer model requires treatment of recipient mice with bleomycin, the collagen produced by transferred cells is overwhelmed by the collagen produced by endogenous lung cells. We nonetheless tried the experiment the reviewer proposed, but as we expected, with our current transfer numbers, we could not detect significant increases in hydroxyproline content by addition of any transferred cell population above the effects of bleomycin alone. Due to the limitation of mouse colony, frequency of cluster 8 cells, and sorting speed, the maximum number of cluster 8 cells we can transfer is currently 3×10^5 cells / mouse in experiments with $n = 4 - 5$. Among the transferred cells, only 10 – 20% can engraft in the host lungs. In contrast, there are approximately 1×10^7 collagen-producing cells in the adult lung (estimated by the frequency of Col-GFP+ cells in lung cell suspension). We are currently working on addressing the functional roles of cluster 8 cells by generating knockin mice, by which we can specifically target cluster 8 cells. However, completion of these studies will take at least another year and we are concerned that waiting this long will affect the timeliness of this paper.

Reviewer #3 (Remarks to the Author):

In this manuscript, the authors described the scRNA-seq analysis using the mouse model of bleomycin-induced lung fibrosis. They particularly focused on Col1a1+ mesenchymal cells and classified them into 12 clusters. They further conducted the PLISH analysis, which is an RNA in situ hybridization method, for detecting the physical localization of the cell population of each cluster. By the series of the analyses, the authors concluded that Cthrc1+ cells (Cluster 8), which presumably descend from alveolar fibroblasts, are pathogenic in bleomycin-induced lung fibrosis. They also conducted the similar scRNA-seq analysis and found that human CTHRC1+ cells are specific to fibrotic lungs in patients with idiopathic pulmonary fibrosis (IPF) and scleroderma. They further characterized Cthrc1+ cells in migration assays in vitro and intratracheal transfer experiments in vivo, demonstrating that Cthrc1+ cells are highly migratory and invasive after intra tracheal transfer to bleomycin-induced lung fibrosis mice.

Overall, I admit this is a high quality paper. However, I'm afraid I have to suggest that the idea of Cthrc1 as a biomarker somewhat lacks novelty as in the light of the previous publications of scRNA-seq using the same model. Moreover, the results from the previous papers seem not always consistent with those described in the present paper. Therefore, I consider further careful extensive analyses should be needed. We thank reviewer #3 for the feedback. We agree that the previous version of our manuscript fell short of addressing some discrepancies with previous publications. We have added new text and figures to address the discrepancies (described below). As reviewer #3 suggested, Cthrc1 has been known to be upregulated in lung fibrosis as reported in Tsukui et al. Am J Pathol 2013 (mouse bleomycin model, ref. 39) or Bauer et al. AJRCMB 2015 (IPF patients, ref. 17) as well as Peyser et al AJRCMB 2019. Therefore, we have never suggested the idea of Cthrc1 as a biomarker in our manuscript. Rather, we think the novelty in our paper is characterizing several distinct populations of collagen-producing cells in the lung, including a distinct, relatively small population that is highly enriched for production of several ECM proteins and marked by expression of CTHRC1 (among other markers). The revised manuscript also now shows that CTHRC1+ fibroblasts are localized in fibroblastic foci in IPF lungs, putting them in the right place to be major players

in pathologic fibrosis. We believe these findings will be useful in advancing lung fibrosis research and can contribute to the community.

General:

At least two papers, in my knowledge, have already reported the scRNA-seq of the bleomycin-induced lung fibrosis mice (Xie, et al. Cell Rep, 2018; Peyser, et al. Am J Respir Cell Mol Biol, 2019). These two studies have also reported the heterogeneity of the activated fibroblasts. Particularly, Peyser et al. have described that *Cthrc1* should be elevated in the activated fibroblasts as well as other biomarkers. However, they concluded that pathogenic fibroblasts could not be defined as a uniquely population. On the other hand, the present study identified that a unique population of *Cthrc1*⁺ cells (Cluster 8) are more pathogenic than the other fibroblasts. To judge which claims should be rationalized, I'm not convinced that the evidence presented in this paper should be sufficiently strong, considering the presented experiments are mostly observational (Figures 1-5). The migration assays in vitro and intratracheal transfer experiments should be the key.

However, they are not conclusive

how their *Cthrc1*⁺ cells affect bleomycin-induced lung fibrosis and human lung fibrosis.

Generally, I'd like to suggest that the present study should put a more focus on further detailed study of the human fibroblasts of the lung fibrosis patients. I'd be particularly interested in seeing further proof-of-concept study, such as contemplating a therapeutic strategy which may target *Cthrc1*⁺ cells (Cluster 8).

Reviewer #3 suggested that our findings had some discrepancies particularly with Peyser et al. AJRCMB 2019. We appreciate reviewer #3 for bringing our attention to this point. We analyzed the data sets of Peyser et al., and found that a fraction of fibroblasts highly expressed *Cthrc1* and other ECM-related genes, which is similar to the cluster 8 cells in our manuscript. We added this analysis to the revised manuscript (pp11 line 202, Supplementary Fig. 4c). There might be multiple reasons that Peyser et al. ended up drawing different conclusions to ours. First, the number of fibroblasts Peyser et al. obtained (1945 cells) is much smaller than ours (12855 cells). It is known that identifying minor populations highly depends on the number of total cells in scRNA-seq (ref. 21). It is possible that *Cthrc1*⁺ fibroblasts did not form a distinct cluster due to the small cell number in Peyser et al.. Second, Peyser et al. performed hierarchical clustering or assigned fibroblasts a signature score by using the "genes which are already known to be upregulated in lung fibrosis". Since those gene lists originated from bulk RNA-seq, which is usually a mixture of a variety of cells, those analyses might undermine the previously unappreciated fibroblast heterogeneity identified by unsupervised clustering. Third, there is no standard dissociation protocol and we have found that the populations obtained highly depend on the dissociation protocol. This could be the major reason that populations in each scRNA-seq study are somewhat different. We are reassured that our new analysis of our collaborators' dataset with a much larger group of patients, confirming a cluster of *CTHRC1*⁺ cells that also express high levels of the ECM protein genes we found in our initial analysis.

We agree that therapeutic experiments targeting *Cthrc1*⁺ fibroblasts should be performed to prove the essential role of *Cthrc1*⁺ fibroblasts in lung fibrosis. Although we are in the process of preparing these genetically modified mice for targeting *Cthrc1*⁺ fibroblasts, it will likely take another year to complete the experiments. Considering the rapid spread of scRNA-seq studies, we believe that we can maximize the value of our findings by publishing soon.

Major comments:

1. The authors should have compared the results with previous studies in more details. They attempted to explore the role of *Cthrc1*⁺ cells but I'd like to suggest that the present data is insufficient to argue whether *Cthrc1* suppress or exacerbate the collagen accumulation in bleomycin-induced lung fibrosis and human

lung fibrosis. Further enriched data supporting the consistency between mouse and human CHTRC1+ fibroblasts should be also needed.

We added the comparison to Peyser et al. as described above. We also added the comparison to previous human scRNA-seq studies as reviewer #1, #2 suggested.

Although we added panels showing CTHRC1 localization at fibroblastic foci in IPF lungs in the revised manuscript (Fig. 5h), we agree that our current data are insufficient to prove the fibrogenic role of Cthrc1⁺ fibroblasts. Therefore, we amended the text to avoid overstatements pp 16 line 332.

2. I'm concerned that the process of the scRNA-seq data should be somewhat arbitrary. For example, the authors employed the t-SNE method for initial clustering and the UMAP method for the second clustering. For the mouse cells, they used the CCA method for the batch-effect correction, while, for the humans, the MNN method was used, which have different characteristics. Especially, please carefully check the results between mice and humans should be compared, nevertheless.

We appreciate the suggestion. We changed all the tSNE plots to UMAP plots in the revised manuscript. We used MNN batch correction for human samples because MNN outperforms CCA when there are more batch-to-batch differences. We added the rationale for using MNN for human samples (pp 11, line 219). We also performed MNN for mouse samples to check the consistency with CCA, and found that MNN batch correction was able to identify the same heterogeneity we found with CCA batch correction for mouse samples (Figure. R1). Thus, we left CCA correction in mice as it was. We also modified the text and figure about identification of COL1A1⁺ clusters in human to clarify the consistency of mouse and human clusters (pp 12 line 225 - 233, Fig. 5e)

Figure R1. UMAP plot of MNN-corrected mouse Col1a1⁺ cells. GFP⁺ samples were aggregated with MNN batch correction and Col1a1⁺ cells were extracted as described in the method. The clusters were annotated by analyzing representative markers of each subpopulation (Fig. 2a).

3. Molecular mechanisms underlying the observed phenotypic appearance of the Cthrc1 cells should be further explored. Otherwise, it would be difficult to identify these cells as the truly important cellular populations in the human fibrosis.

We performed in vitro stimulation of primary alveolar fibroblasts with TGF- β , TNF- α , and EGF in the revised manuscript. We found that TGF- β stimulation increased expression of several cluster 8 markers, including Col1a1, Cthrc1, Postn, and Tnc. Those genes were not upregulated by TNF- α or EGF. These

results suggest that TGF- β may be one of the inducers of cluster 8 cells in alveolar fibroblasts. This result is shown in Supplementary Fig. 7c, pp 15, line 295.

Minor comments:

4. Figure 1: For the scRNA-seq, I'm not convinced that the identified $Col1a1^+$ cells should represent the entire population of collagen producing fibroblasts. The authors stated "we sorted GFP $^-$ cells to compare gene expression patterns in the cells responsible for collagen production to pattern seen in other cell types," I could not find the data. Please further explain the rationale in analyzing the $Col1a1^+$ cells over the other pan-fibroblast markers such as, CD90 and Vimentin, in order to avoid the selection biases.

We appreciate the feedback. We added a new panel (Fig. 1b), which is color-coded by GFP $^+$ and GFP $^-$ samples. This panel showed that $Col1a1^+$ cells shown in Fig. 1c are mostly from the GFP $^+$ samples. Although there are some cells from GFP $^+$ samples in lineage $^+$ clusters such as endothelial and epithelial cells, we think these can be explained by contamination of lineage $^+$ cells in the GFP $^+$ gate at the time of sorting. Fig. 1c also showed that $Col1a1$ expression is mostly restricted to the cells we focused on in the following analyses. The clusters we removed for subsequent re-clustering and analysis of $Col1a1^+$ cells expressed lineage markers that clearly identified other known cells types and, with the exception of the very small cluster identified as mesothelial cells, did not consistently express $Col1a1$ (or $Col1a2$ or $Col3a1$). Some studies used Thy1 or Vimentin as fibroblast markers. However, we are not aware of studies which validated these markers as pan-fibroblast markers. In our data, Thy1 was weakly expressed in adventitial fibroblasts but not in other fibroblasts in mouse (Figure R2). T cells had much higher expression of Thy1 (Figure R2). Vimentin was expressed in almost all lung cells except epithelial cells (Figure R2). Several other markers that have previously be used to identify subsets of tissue fibroblasts (such as *Pdgfra* and *Tcf21*, did appear to be captured by the approach we used here, as we now show in Supplementary Fig 1a. We agree that there may be other stromal cells in normal and/or fibrotic lungs that express levels of collagen not detected by our sequencing method and that these cells might have the capacity to differentiate into collagen-producing cells. However, we do not have any good marker to identify such cells. Future lineage tracing experiments with tools we are currently developing to mark each of the collagen producing subsets identified by the current work could allow us to determine if there are important precursors we are missing, but these studies are clearly beyond the scope of the current paper. Collagen 1 is known as the most abundant ECM in tissue. Thus, we believe that focusing on $Col1a1^+$ cells is a simple but effective way to study collagen-producing cells.

Figure R2. UMAP plots of all lung cells overlaid with gene expressions. The third annotated panel is from Fig. 1c. See Supplementary Fig. 1a for lineage identification.

5. Generally, the number of cells are described as “approximately”. Perhaps in Supplementary File, please provide the exact number how many cells belong to each of the clusters. If the authors employ different methods or the parameters for each dataset, the robustness of the results should be evaluated in a quantitative manner.

We omitted “approximately” and amended the numbers to the exact numbers. We added tables showing cell numbers of each cluster in Supplementary Fig. 1b for mouse and Supplementary Fig. 5b for human. As described in Figure R1 above, we confirmed that CCA batch correction yielded the same conclusions as MNN batch correction in mouse.

6. pp 9 and Figure 4: The authors suggested that the *Coll1a1*⁺ cells may be derived from alveolar fibroblasts based on the results of RNA velocity and pseudotime trajectory analyses. I wonder if the authors could actually differentiate the *Coll1a1*⁺ cells into the *Cthrc1*⁺ cells, for example, by *TGFB1* or any other fibrosis-related cytokine treatments. *TGFb* stimulation can sometimes convert the lipofibroblasts to myofibroblasts (Agha, et al. *Cell Stem Cell*, 2017).

As described above, we added a new panel, in which we stimulated primary alveolar fibroblasts with *TGFb*, *TNF-a*, and *EGF* (Supplementary Fig. 7c).

7. The data of human *CTHRC1*⁺ fibroblasts are just an observation study, which should become more convincing to ensure the consistency between the mouse *Cthrc1*⁺ cells in Cluster 8 and the human *CTHRC1*⁺ fibroblasts. For example, a similar immunostaining analysis or PLISH analysis for *CTHRC1* in the human specimen of IPF should be needed. Positional relationship between the human *CTHRC1*⁺ cells and the myofibroblasts is particularly interesting.

We performed in situ hybridization and antibody staining of *CTHRC1* in IPF lungs, and found that *CTHRC1*⁺ fibroblasts were localized within fibroblastic foci. We added these panels to Fig. 5h.

8. Figure 6B: Violin plots of each marker, as shown in Figures 2A and 4A, would be helpful to understand the results in a more quantitative manner.

As the reviewer suggests, we have replaced our table with violin plots in Figure 6B.

9. Figure 6C: The data clearly represents the presence of two populations by the gating of “Alveolar”; one is *CD9*^{high} and the other is *CD9*^{low}. What is the *CD9*^{high} *PDGFR*⁺ population? Similarly, in the last panel of Figures 6C and 7C, there is a *CD9* negative population. What is that population?

We thank the reviewer for pointing out the presence of subsets of cells from our flow sorting strategy that were not included in our transfer experiments. One limitation of flow sorting is that the harsh digestion conditions needed to extract all of the collagen-producing cells from fibrotic tissue can lead to loss of some cell surface markers. We suspect that some loss of *PDGFRa* expression as a consequence of digestion might explain the appearance of a large group of *CD9* negative cells in cells sorted in the *PDGFRa*⁻ gate. We acknowledge that the *PDGFRa*⁺ population does include some *CD9* high cells, which is consistent with the scattered *CD9*⁺ cells shown within the alveolar fibroblast clusters in Fig. 6a. These cells did not group together as a uniform cluster. The goal of our sorting strategy was to broadly separate alveolar fibroblasts, adventitial fibroblasts, peribronchial fibroblasts, smooth muscle cells/pericytes, and *CTHRC1*⁺ cells (cluster 8). Our qPCR findings suggest that this strategy was largely successful.

10. It is preferable to include the migratory assay for the human *CTHRC1*⁺ fibroblasts as well

We agree that the migratory assay for the human CTHRC1⁺ fibroblasts will help us understand their functional role. However, we are still working hard to optimize methods to purify this population from human lungs, a process that is limited by the effects of protease digestion conditions on expression of each of the relevant cell surface markers and the inconsistent availability of fibrotic human lung samples to work with. We expect that this process will take several more months and are concerned that waiting to perfect these methods would compromise the timeliness of our current work.

11. Figure 7E: How would the Cluster 8 cells affect the bleomycin-induced lung fibrosis? Would they suppress or exacerbate the fibrosis? The role of Cthrc1 in the Cluster 8 cells should be more clearly addressed.

As we described to reviewer #2, assessing collagen deposition from transferred cluster 8 cells is technically challenging. We are working on generating genetically modified mice to target cluster 8 cells, but we would like to present these experiments in our next study because it may take another year. Regarding the role of Cthrc1 in cluster 8 cells, it is known that Cthrc1 enhances the migration capacity of fibroblasts or invasiveness of cancer cells. It is possible that Cthrc1 in cluster 8 cells also enhances migratory and invasive capacity of cluster 8 cells. We discussed these possible roles of Cthrc1 starting pp 18, line 375. However, we are not trying to claim that CTHRC1 itself plays a role in causing tissue fibrosis. In fact, two previous studies using Cthrc1 knockout mice report conflicting results, one suggesting that Cthrc1 is protective and the other that it contributes to fibrosis. In this paper, we were more interested in using Cthrc1 as a marker of population of cells that we think deserve further study as potential drivers of pulmonary fibrosis.

12. Figures 7C, D, E and F: What are the indications obtained from the results of the peribronchial fibroblasts and Mcam+ cells.

In Fig. 7c, d, e, we chose alveolar and adventitial fibroblasts as comparison to cluster 8 cells because alveolar and adventitial fibroblasts were the two populations with the greatest capacity to colonize bleomycin treated lungs (Fig. 6f). We subsequently also performed migration assays with alveolar, adventitial, peribronchial, and Mcam+ cells isolated from untreated lungs (Figure R3), but none of these cells showed remarkable migration capacity when they were collected from untreated lungs; Mcam+ cells showed the least migration, but the migration capacity of all of these cells (max 2% over 48 hours) was quite minimal. Because there are already 7 main and 8 supplementary figures, we decided that these results were not impactful enough to share in the manuscript.

Figure R3. Migration assay with the cells isolated from untreated lungs.

Miscellaneous comments:

13. pp7, line 146: Please correct “Figure 2D” to “Figure 3D”

Thank you. We corrected this typographical error in the revised manuscript.

14. pp9, line 186-192: The authors address that cluster 0 were differentiated into clusters 1 and 2, then cluster 2 differentiated into cluster 8. Considering that the RNA velocity data (Figure 4D), cluster 1 seems to be converted to cluster 0 and 2, then cluster 2 turned into cluster 8. Is that right?

We are a bit more conservative in interpreting the RNA velocity data. In our view, and the view of our bioinformatics consultants, there is no coherent assembly of arrows from cluster 0 to cluster 1 or vice versa. We think the only consistent directionality of arrows with this method is from cluster 2 to cluster 8. Our pseudotime analysis does suggest differentiation of cluster 0 to 1 and 2 and cluster 2 to 8. As in the text, we acknowledge that these computational methods are really only a way to generate hypotheses, but that rigorous testing will require the generation of better tools for lineage tracing, which we are actively working to develop.

Reviewers' comments:

Reviewer #1 (Remarks to the Author):

The manuscript has been considerably strengthened following addition of validating data from the Yale cohort. I have no additional comments to make on the revision. This is a nicely written and very interesting study for which the authors are to be congratulated.

Toby Maher

Reviewer #2 (Remarks to the Author):

The authors have responded appropriately to my critique.

Peter Bitterman

Reviewer #3 (Remarks to the Author):

I appreciate the authors' responses and especially their extensive analyses. I agree that the manuscript has been improved with the enriched data and the deepened discussion. Nevertheless, I found this paper has not yet essentially characterized the roles of Cthrc1 in cluster 8 cells or the features of human Ctrc1+ fibroblasts in details. On the other hand, they have conducted re-evaluation of the previously published datasets, recruiting the substantial amount of data from normal individuals and IPF patients, as a new collaboration, for which I truly appreciate. Overall, even though Cthrc1 itself may not be a novel marker of pulmonary fibrosis, strictly speaking, it seems that Cthrc1+ cells, other than myofibroblasts, should play an important role in pulmonary fibrosis. Indeed, it is true that the significance of Cthrc1+ cells is now properly demonstrated by their data, at least in the mouse model. However, further careful consideration should be needed when we confidently apply the obtained knowledge to human IPFs. To this goal, I consider this manuscript still contains an important flaw, leaving an impression that the presented data still remains mostly descriptive, even after some human transcriptome data has been added.

Comments to Figure 5h:

1. Did the authors observe any relation between Cthrc+ cells and Acta2+ cells in their localizations? I wonder if those two cellular populations are involved in the same fibrotic focus or in separate foci? Double immunostaining is generally recommended.

2. How frequent Cthrc+ cells are found in fibrotic foci? The presented data depicts that Cthrc+ cells are new component cells in fibrotic foci, although fibrotic foci are generally known to involve Acta2+ myofibroblasts. My concern is that the authors just show only one "fibrotic focus" from one IPF patient. Please include a figure to represent the observatio in a more general manner.

Response to reviewers' comments:

Reviewer #1 (Remarks to the Author):

The manuscript has been considerably strengthened following addition of validating data from the Yale cohort. I have no additional comments to make on the revision. This is a nicely written and very interesting study for which the authors are to be congratulated.

Toby Maher

Reviewer #2 (Remarks to the Author):

The authors have responded appropriately to my critique.

Peter Bitterman

Reviewer #3 (Remarks to the Author):

I appreciate the authors' responses and especially their extensive analyses. I agree that the manuscript has been improved with the enriched data and the deepened discussion. Nevertheless, I found this paper has not yet essentially characterized the roles of *Cthrc1* in cluster 8 cells or the features of human *Ctrc1+* fibroblasts in details. On the other hand, they have conducted re-evaluation of the previously published datasets, recruiting the substantial amount of data from normal individuals and IPF patients, as a new collaboration, for which I truly appreciate.

Overall, even though *Cthrc1* itself may not be a novel marker of pulmonary fibrosis, strictly speaking, it seems that *Cthrc1+* cells, other than myofibroblasts, should play an important role in pulmonary fibrosis. Indeed, it is true that the significance of *Cthrc1+* cells is now properly demonstrated by their data, at least in the mouse model. However, further careful consideration should be needed when we confidently apply the obtained knowledge to human IPFs. To this goal, I consider this manuscript still contains an important flaw, leaving an impression that the presented data still remains mostly descriptive, even after some human transcriptome data has been added.

We thank reviewer #3 for the thoughtful review. We would like to point out that we have never suggested that *Cthrc1+* pathologic fibroblasts do not express α -SMA. Indeed, our new data show that *CTHRC1+* fibroblasts in fibroblastic foci of IPF are most likely a subset of the cells others have described based on α -SMA staining (Fig. 6b). Our manuscript also did mention that *Cthrc1+* pathologic fibroblasts up-regulate *Acta2* compared to their putative progenitors in scRNA-seq (pp 18, line 379-388). Although α -SMA can be one of the activation markers of fibroblasts as a large body of literature shows, our data suggest that α -SMA by itself cannot identify pathologic fibroblasts at least in the lung. Since myofibroblasts are often identified only by α -SMA staining, the current definition of myofibroblasts is unable to distinguish *CTHRC1+* pathologic fibroblasts from other α -SMA+ cells.

In addition to the figure modification, we amended a sentence in Introduction to avoid the confusion regarding myofibroblasts as follows.

(pp 4, line 60)

Before: However, previous work by ourselves and others suggest that most collagen-producing cells in normal murine lungs and in murine models of pulmonary fibrosis express little or no α -SMA.

After: However, previous work by ourselves and others suggest that α -SMA is an inconsistent marker of collagen-producing cells in normal murine lungs and in murine models of pulmonary fibrosis.

Comments to Figure 5h:

1. Did the authors observe any relation between Cthrc+ cells and Acta2+ cells in their localizations? I wonder if those two cellular populations are involved in the same fibrotic focus or in separate foci? Double immunostaining is generally recommended.

We added images of CTHRC1 and α -SMA staining in sequential sections of IPF lungs (Fig. 6b). Fibroblastic foci in IPF contained both α -SMA and CTHRC1+ cells, but not all α -SMA+ cells were CTHRC1+. Because of the limitations of serial sectioning, we could not establish with certainty whether or not all of the CTHRC1+ cells were also positive for α -SMA staining. As the modified text now clarifies (pp 13, line 267-270), however, α -SMA is not specific to fibroblastic foci. These data suggest that α -SMA immunostaining by itself cannot identify pathologic fibroblasts.

2. How frequent Cthrc+ cells are found in fibrotic foci? The presented data depicts that Cthrc+ cells are new component cells in fibrotic foci, although fibrotic foci are generally known to involve Acta2+ myofibroblasts. My concern is that the authors just show only one “fibrotic focus” from one IPF patient. Please include a figure to represent the observation in a more general manner.

We appreciate the suggestion. As the reviewer suggested, we confirmed that fibroblastic foci are α -SMA+ as shown in previous reports (Fig. 6b). All the fibroblastic foci we identified included CTHRC1+ cells. In Fig. 6a, we added images from 3 IPF patients to show the reproducibility of CTHRC1 staining in fibroblastic foci. Fig. 6b includes representative images from 4 IPF patients (different patients from Fig. 6a), as described in the figure legend.

REVIEWERS' COMMENTS:

Reviewer #3 (Remarks to the Author):

I sincerely appreciate the extensive analyses of the authors and the thorough explanations of my previous concerns. Indeed, I apologize some misunderstandings in my previous comments, which have been completely addressed now.